

# The effect of tip-speed ratio and free-stream turbulence on the coupled wind turbine blade/wake dynamics

Francisco J. G. de Oliveira[1], Martin Bourhis[1], Zahra Sharif Khodaei[1], and Oliver R. H. Buxton[1]

[1]Department of Aeronautics, Imperial College London, UK

**Correspondence:** Francisco J. G. de Oliveira (f.oliveira22@imperial.ac.uk)

**Abstract.** Wind turbines operating within wind farms experience complex aerodynamic loading arising from the interplay between wake-induced velocity deficits, enhanced turbulence, and varying operational conditions. Understanding the relationship between the blade's structural response to the different operating regimes and flow structures generated in the turbine's wake is critical for predicting fatigue damage and optimizing turbine performance. In this work, we implement a novel technique, allowing us to simultaneously measure spatially distributed blade strain and wake dynamics for a model wind turbine under controlled free-stream turbulence (FST) and tip-speed ratio ($\lambda$) conditions. A $1\,\mathrm{m}$ diameter three-bladed rotor was instrumented with distributed Rayleigh backscattering fibre-optic sensors, while synchronised hot-wire anemometry captured wake evolution up to $4$ rotor diameters downstream. Experiments were conducted covering a wide $\{\mathrm{FST}, \lambda\}$ parameter space—21 cases in total. Results reveal that aerodynamic-induced strain fluctuations peak at $\lambda \approx 3.5$, close to the design tip -speed ratio ($\lambda_d = 4$), with the blade's tip experiencing a contribution from the aerodynamically-driven strain fluctuations of up to $75\%$ of the total fluctuating strain at design conditions. Spectral analysis shows frequency-selective coupling between wake flow structures and the blade response, dominated by flow structures dynamically related to the rotor's rotating frequency (*eg.* tip vortex structure). The novel experimental methodology and results establish a data-driven foundation for future aeroelastic models' validation, and fatigue-informed control strategies.

## 1 Introduction

Wind energy dominates current global decarbonization strategies, and is projected to be one of the largest contributors to renewable electricity generation in the upcoming decades (Veers et al., 2019). To meet rising market demand for wind energy, wind turbines are typically installed as wind farms which typically arrange the turbines as compact arrays, as wind farm commissioners seek to maximise the wind power extraction within limited lease areas, whilst minimising costs associated with foundations, cabling, and infrastructure. As a result, any turbine behind the front row receives airflow that is a blend of upstream wakes and atmospheric turbulence (Stevens and Meneveau, 2017; Porte-Agel et al., 2020). These are then exposed





to an incoming flow characterised by velocity deficits and enhanced turbulence, introducing strong spatiotemporal variability

into the operational environment of turbines within a wind farm (Barthelmie et al., 2009; Stevens and Meneveau, 2017; Porte-Agel et al., 2020; Bossuyt et al., 2023). This comes at the cost of power deficits across the span of the wind farm operation (Scott et al., 2024), and increased fatigue loading experienced by the machines (Thomsen and Sørensen, 1999; McGugan and Mishnaevsky, 2020; Pacheco et al., 2024).

Amongst wind turbine components, blades are one of the most sensitive components to unsteady aerodynamic loading

regimes, (Ismaiel and Yosida, 2018). Wind turbine blades are exposed to centrifugal, gravitational and aerodynamic forces (Wen et al., 2020; Jenkins et al., 2021). The experienced gravitational and centrifugal forces are determined by the blade structure, mass distribution, and tip-speed ratio ($\lambda$), and result in cyclic blade loading (Wen et al., 2020). The last encompasses all the aerodynamic and atmospheric effects to which a turbine within a wind farm is exposed:

1. the sectional lift and drag forces the blade experiences, accumulated throughout the blade (Jenkins et al., 2021);

2. the influence of free-stream turbulence (FST) impacting on the blade via direct and indirect effects (de Oliveira et al., 2025), where the first refers to the direct imposition of a turbulent buffeting on the surface of the blade, the latter to the modification of vortex shedding mechanisms of the blade, which subsequently imprint themselves into the mechanical response of the blade;

3. impact of wind gusts and flow stratification across the turbine's sweep area (Kwon et al., 2012; Chamorro et al., 2015);

4. turbine-to-turbine wake effects, where a blade can be exposed to both waked and non-waked conditions (Thomsen and Sørensen, 1999; Hansen et al., 2014; McGugan and Mishnaevsky, 2020; Pacheco et al., 2024).

Modern wind turbines are far larger and more flexible than previous designs. Wind characteristics that were negligible for smaller and more rigid wind turbines may now significantly impact the structural dynamics of modern turbines (Moreno et al., 2025a). Monitoring and assessing the impact of operating and FST conditions on the structural integrity of wind turbines thus

becomes of the utmost relevance in implementing blade-loading controlled strategies. In this work, we will focus on effects 1 and 2.

A key parameter influencing both wake dynamics and blade loading is the tip-speed ratio ($\lambda$), defined as the ratio between the blade tip speed and the time-averaged incoming wind velocity ($\lambda = R\Omega/U_\infty$, where $R$ corresponds to the wind turbine radius, $\Omega$ to the turbine rotational speed). The effect of $\lambda$ on the turbine's blade loading mechanisms is twofold: at a fixed

wind speed, $\lambda$ sets the rotor's angular velocity and, consequently, the period of the cyclic gravitational and centrifugal blade loading; and, $\lambda$ also determines the rotor's aerodynamic loading—the lift and drag acting on the turbine blades—and thus the efficiency of the aeromechanical power conversion (Bastankhah and Porté-Agel, 2017). Therefore, $\lambda$ is the major parameter contributing to the impact of effect 1. Moreover, $\lambda$ governs the strength and coherence of the flow structures shed by the turbine and present in its wake (Hansen, 2015; Biswas and Buxton, 2024a). Below the design tip-speed ratio ($\lambda_d$), blades operate in a

stalled regime dominated by flow separation and low lift generation, while at high $\lambda$, the tip region may become unloaded due to reduced angles of attack, altering both power output and structural loading patterns. This results in spatially non-uniform





load dynamics along the blade, with different regions (*e.g.* the root, mid-span, tip) responding to distinct locally generated flow-structures that depend on the inflow conditions and operating regime (Biswas and Buxton, 2024a). Moreover, the instantaneous inflow velocity $U_\infty$ varies continuously over the turbine's service life (Moreno et al., 2025b), and modern control schemes strive to track an optimal $\lambda$ (*i.e.*, associated to maximum wind power extraction). However, during transients—whether due to gusts, start-up/shutdown sequences, or deliberate off-design operation—$\lambda$ departs from its optimum setpoint, producing time-dependent deviations in aerodynamic and structural loading. Consequently, quantifying the sensitivity of loading regimes of varying $\lambda(t)$ under realistic, unsteady inflow conditions is critical for reliable fatigue and ultimate-strength assessments.

The interplay between FST and $\lambda$ introduces complex variations in wake structure and blade response. FST is largely associated in literature with a decreased lifetime of wind turbines (Thomsen and Sørensen, 1999; Ismaiel and Yosida, 2018; McGugan and Mishnaevsky, 2020; Pacheco et al., 2024). FST impacts not only the structure, but the flow development of the wake of a turbine. Increased FST has been associated with weaker tip vortices that break down closer to the turbine, leading to more isotropic and broadband turbulence in the wake of a turbine (Gambuzza and Ganapathisubramani, 2023; Biswas and Buxton, 2025; Bourhis et al., 2025). The presence of FST increases the energy of near-wake coherent structures shed by bluff bodies, which in turn amplifies their impact on the structural dynamics of the body (Maryami et al., 2020; de Oliveira et al., 2025). In aerofoil profiles, FST delays stall and promotes the lift/drag coefficient (Maldonado et al., 2015; Thompson et al., 2023). Meanwhile, the presence of FST has been associated with an increased power coefficient of wind turbine models (Medici and Alfredsson, 2006; Gambuzza and Ganapathisubramani, 2021, 2023). Understanding spatially distributed loading along the blade span is essential to determine how coherent flow structures like tip vortices translate into blade response, ultimately leading to fatigue damage; and how the different operating conditions model the blade's sectional loading. To the author's knowledge, literature lacks attempts to build correlations between wake dynamics, influenced by both FST and $\lambda$, to structural events experienced across different sections of the blade. Spatially distributed structural information across the blade's span can be exploited to correlate the turbine's wake dynamics with the blade's structural response.

Traditional instrumentation using load cells at the hub, single-point strain gauges, fibre-Bragg gratings (FBGs), pressure taps or transducers, either integrate loads over the entire blade, or require cumbersome plumbing for each measurement port, limiting the achievable spatial resolution and complicating blade attachment, altering blade stiffness and profile while attempting to acquire spatially refined information of the structure (Campagnolo, 2013; Wen et al., 2020; Pacheco et al., 2024; de Oliveira et al., 2024). To resolve these spanwise variations in unsteady loading mechanisms, we employ distributed Rayleigh backscattering sensors (RBS). These sensors overcome these challenges by leveraging single-mode optical fibers that can be adhesively bonded along a blade without appreciable mass, stiffness and modification-of-the-surrounding-flow penalties, while capturing a dense spatial network of data-points (de Oliveira et al., 2024; Li and Sharif-Khodaei, 2025; Zhou et al., 1999). An optical-frequency-domain-reflectometry (OFDR) system interrogates the random Rayleigh backscatter arising from micro-modulations in the fiber's refractive index, induced by the local strain field (Li and Sharif-Khodaei, 2025; Xu and Khodaei, 2020). This approach delivers sensing points every $2.6\,\mathrm{mm}$ along the sensing area instrumented, using a single optical channel, providing a continuous map of strain that can be correlated to local aerodynamic loads. Crucially, by integrating an RBS-equipped blade with simultaneous wake-measurements, we assess in space and time how the multitude of flow scales evolving



in the wake of a turbine are imprinted on the blade dynamics, along its span. The blade dynamics retrieved from the fibre-optic sensors are passed through the rotor via a fibre-optic slip-ring, transmitting the rotating signal to the stationary interrogation unit. Similar devices have been used in experimental setups to instrument blades of wind turbine models with FBGs in Wen et al. (2020); Campagnolo (2013); Shroff and *et. al.* (2017). The spatio-temporal coupling of concurrent measurements of spatially resolved structural dynamics and wake dynamics is key to identifying flow induced fatigue hotspots, and to produce correlations between flow dynamics and induced blade deformation/loads (de Oliveira et al., 2024, 2025), that will help guide wind turbine control methodologies on minimization of fatigue damage, while maximizing power production per wind turbine.

In this study, we present simultaneous measurements of the flow field behind a $D = 1\text{m}$ wind turbine model, and the distributed strain response across one of the wind turbine's blades, across a range of $\lambda$ and FST conditions. By combining hot-wire anemometry with RBS, we aim to assess how the interplay between $\lambda$ and FST modulates blade loading and respective wake structure, providing key knowledge on the underlying fluid-structure interaction mechanisms that govern turbine performance in realistic operating conditions. For the first time, we present both dynamically and spatially resolved strain measurements along a turbine model blade's span, and concurrent wake velocity measurements.

## 2 Experimental Methodology

A large number of simultaneous RBS and hot-wire anemometry experiments were conducted in the large test section of the closed-loop $10 \times 5$ wind tunnel in the Department of Aeronautics at Imperial College London. The cross section of the test section is $5.7 \times 2.8\text{m}^2$, spanning for $18\text{m}$. The wind speed and temperature are controlled via a closed-loop control system. A passive regular turbulence-generating grid with a mesh size $M = 110\text{mm}$ was placed upstream of the turbine model, generating the FST to which the turbine model was exposed. Different cases of FST were produced by adjusting the grid-to-turbine distance. Preliminary hot-wire anemometry experiments quantified the decay of the turbulence generated by the grid over a large set of streamwise distances, along the centreline of the tunnel, and hub-height. This allowed us to determine prior to the installation of the turbine the grid-turbine distances to be tested, generating 3 different FST "flavours", denoted by $\{A, B, C\}$. Figure 1 *a)* presents the FST parameter space $\{TI, \mathcal{L}\}$ considered in this study. Cases $\{A, B, C\}$ correspond to grid-turbine distances of, $\Delta x/M \in \{50, 20, 10\}$ respectively. Each of the FST cases is associated with a respective turbulence intensity ($TI$) and integral length scale $\mathcal{L}/R$, defined as:

$$TI = \sqrt{\langle u'^2 \rangle}/U_\infty, \tag{1}$$

and,

$$\mathcal{L}_{11}/R = U_\infty \int\limits_0^{\tau_0} \underbrace{\frac{\langle u'(t)u'(t+\tau)\rangle}{\langle u'^2 \rangle}}_{R(\tau)} \, \mathrm{d}\tau, \tag{2}$$

where $U_\infty$ and $u'$ correspond respectively to the free-stream velocity and streamwise velocity fluctuations, and $\tau_0$ to the first zero crossing of the autocorrelation function ($R(\tau)$) of $u'$. The integral length scale of the FST is also normalised by the

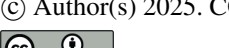



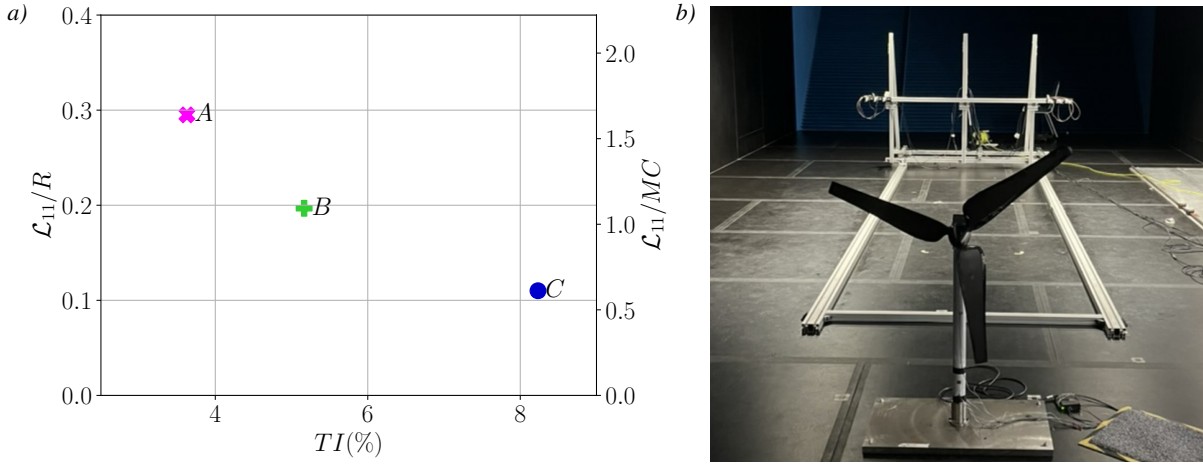

**Figure 1.** *a)*: Characterisation of the free-stream turbulence (FST) conditions used in the experiment. Each case corresponds to a different grid-turbine spacing $x/M$, generating distinct turbulence intensity ($TI$) and integral length scale $\mathcal{L}/R$. Increasing turbulence levels are denoted by cases $A$, $B$, and $C$ with decreasing $x/M$ ratios. Integral length scale is defined both as a ratio of the turbine radius ($R$), and length of the chord at midspan ($MC$). *b)*: picture of the experimental apparatus set at the centre of the $10' \times 5'$ wind tunnel top section.

chord length at the midspan, $MC$. While the turbulence is generated using passive grids, and is therefore often idealized as homogeneous and isotropic, the present study does not aim to investigate such idealized turbulence *per se*. Instead, the three FST cases are selected to span a range of intensities and length scales representative of the parameter space that a wind turbine
may realistically be exposed to.

## 2.1 Wind Turbine Model and Experimental Setup

The wind turbine model consists of a three-bladed rotor with a diameter ($2R$) of 1m, imposing a blockage ratio of 4.9% to the wind tunnel's cross sectional area. A picture of the experimental apparatus set in the wind tunnel is presented in figure 1 *b)*. The spanwise chord length and twist angles were computed using an in-house Blade Element Momentum (BEM) code,
optimised for a tip speed ratio of $\lambda_d = 4$ (Gouder et al., 2024).

    The rotor is mounted on a nacelle through a shaft, sitting on the turbine's tower. The nacelle was machined from a single block of aluminium, resulting in a 150mm long box-shaped nacelle, with a cross section of 60x60mm$^2$ being smaller than the hub of the turbine's rotor (with a diameter of 70mm). The turbine's rotational speed ($\Omega$) is controlled using a perpendicularly mounted DC electric motor, operating as a generator. The system is equipped with an inline optical encoder for accurate
rotational speed measurement. The generator is connected to the rotor shaft through a $1:3$ polyketone gear ratio system. The main shaft housing the rotor rotates along the nacelle on two sets of ball bearings to ensure a smooth rotation. The tower of the turbine is made of a single 800mm long stainless steel tube, with a 50.8mm outer diameter, and internal diameter of $\approx 45$mm to be able to house the generator. The tower is mounted to the bottom of the tunnel with a collar-like connection to a 40mm inch thick steel plate. The turbine was designed to approximate as closely as possible design ratios typically seen in real life





wind turbines, resulting in a nacelle length to turbine diameter ratio of $0.15$, turbine diameter to hub height ratio of $1.25$ and turbine to tower diameter ratio of $20$.

For all experimental conditions, the averaged incoming free-stream wind speed was kept at $U_\infty = 2.8$ m/s via a PID controller connected to a pitot-static tube. The power input to the wind tunnel fans was then adjusted to keep the immediate inflow velocity to the wind turbine constant through all the cases tested. This sets the Reynolds number based on the turbine diam-

eter to be approximately $200,000$, above $Re \approx 9.3 \times 10^4$ after which the main flow statistics (such as mean velocity, velocity skewness and turbulence intensity profiles) become independent of the Reynolds number (Chamorro et al., 2011). Experiments were conducted at 7 tip speed ratios ($\lambda \in \{1, 2, 3, 3.5, 4, 5, 6.5\}$), for each FST "flavour", resulting in 21 combinations tested in the $\{\lambda, \text{FST}\}$ parameter space.

## 2.2    Rayleigh backscattering sensors(RBS) strain measurements

The pressure side of one turbine blade was instrumented with RBS, enabling high-spatial-resolution measurements of the strain distribution along the blade's span. In contrast to FBGs—commonly employed for blade monitoring but limited to localised measurements (Campagnolo, 2013; Shroff and *et. al.*, 2017; Wen et al., 2020)—RBS allow continuous sensing of the blade's shape. The sensors consist of single-mode optical fibers (SMF) with a core diameter of $9\mu m$ and cladding diameter of $125\ \mu m$ (de Oliveira et al., 2024). We have instrumented the pressure side of the blade, as it has been reported to experience the largest

accumulated strain when exposed to free-stream turbulence (Pacheco et al., 2024). The fiber optics were bonded to the blade using cyanoacrylate glue, and were set on a sinusoidal layout following an optimised configuration (see figure 2 *a)*), designed to minimize optical losses (Li and Sharif-Khodaei, 2025). Moreover, the chosen layout allows us to acquire simultaneously comprehensive strain information of the flapwise and edgewise bending mechanisms using axial strain sensors (this is done by decomposing the acquired strain along the blade's cartesian coordinate system $s\mathcal{O}c$ as presented in fig. 2), acting on the blade

along multiple spanwise sections of the blade.

Following Li and Sharif-Khodaei (2025), a reference grid with $a = 40$ mm spacing was used as the foundation for the fiber optic layout, ensuring precise positioning and repeatability across the different blade sections. The final sensor layout used spans for $s/R = [0.15, 0.95]$, and covers $\approx 35\%$ of the available blade chord length at the root, to $\approx 90\%$ of the available blade chord length close to the tip region of the blade. Sensors are separated by $\delta f = 2.6$mm along the fibre optic span, providing

detailed spatial dynamics of the blade's behaviour during the turbine's operation.

A fibre optic slip ring (MFO100) was used to allow the continuous fibre optic strain measurements whilst the turbine was under operation. The slip ring model used is rated for a maximum rotational speed of $2000$rpm, and was fine tuned to work within the wavelength range of operation of the laser source used—Luna Ltd ODiSI-B—at $\lambda_f = 1550$nm, possessing a maximum insertion loss of $1.5$dB. The fibre optic slip ring is set inline with the rotor's driving shaft, mounted at it's root (see the close-up

schematic in fig. 2 *b)*), allowing the optical signal to pass through the rotating assembly, without interruption or signal degradation. The slip ring assembly is integrated into the nacelle design, with the stator fibre optic cable connected to the ODiSI-B interrogation unit (LUNA Ltd.), providing the reference measurements of the fibre optics, and the rotor cable interfacing with the blade-mounted fiber sensors. The slip ring is connected to the sensing path via a FC/APC-FC/APC connector, to minimize





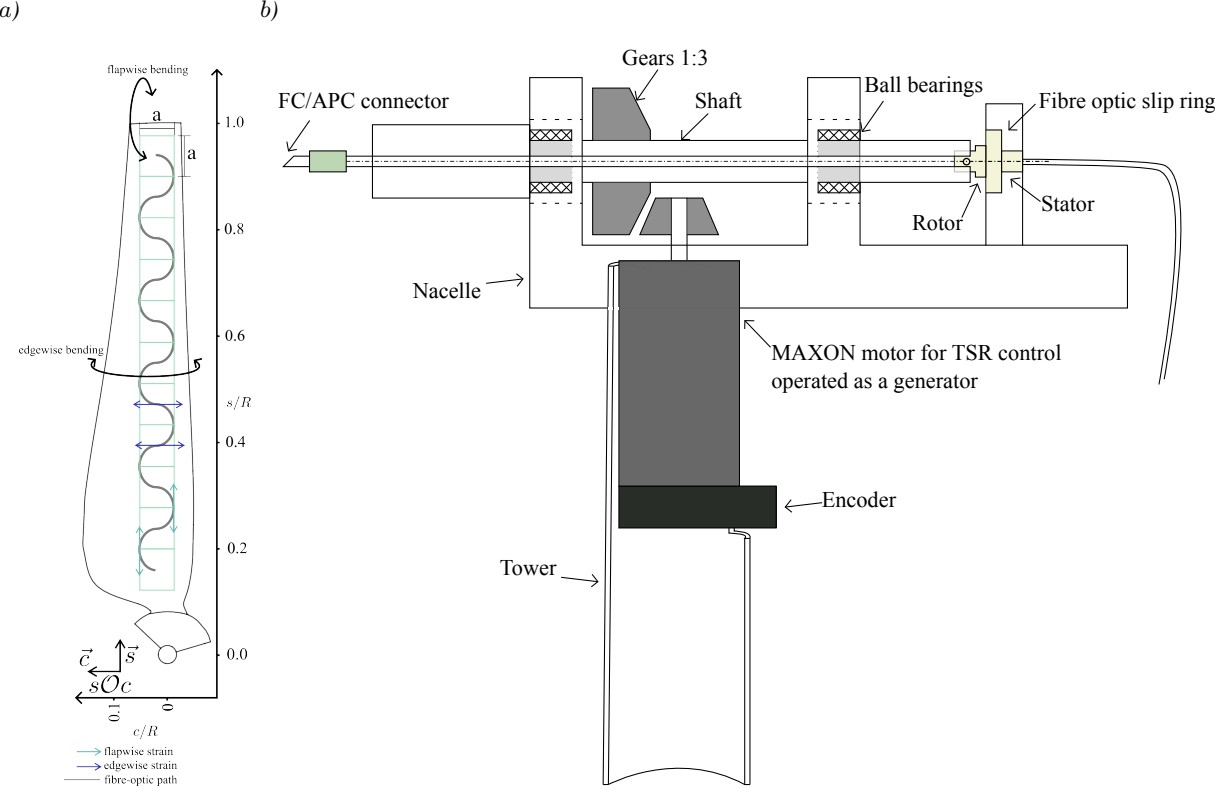

**Figure 2.** (*a*) Schematic of the Rayleigh backscattering sensor layout along the pressure side of the wind turbine blade. The sinusoidal fiber path, defined on a reference grid with spacing $a = 40\,\mathrm{mm}$, allows the retrieval of both the edgewise and flapwise strain components. (*b*) Schematic of the nacelle, where the optical slip ring (MFO100) is mounted on the nacelle, at the base of the rotor shaft, allowing the continuous optical signal transmission from the rotating blade, to the stationary optical interrogator (ODiSI-B).

backscattering at the interface of the connector, while maintaining a secured bolted connection to the slip ring rotor cable. The
RBS data is acquired at the maximum available frequency of 100Hz at the spatial resolution used, a limitation inherent to the coupling between the temporal, and spatial measurement of RBS. Each measurement acquisition provides simultaneous strain readings from all sensor locations along the instrumented blade sections. The wind turbine's diameter and wind speed were chosen to ensure that the maximum tested above-optimum operating conditions ($\lambda > \lambda_d$) resulted in more than $N_{acq}^{ROT} > 15$ acquisitions per rotation (see tab. 1). The RBS measures strain in units of microstrain ($\mu\varepsilon$) and from now on in this work, $\varepsilon$
corresponds to strain measured in $\mu\varepsilon$.

Before and after each set of acquisitions across fixed $\lambda$ and FST conditions, a test was conducted under quiescent background conditions, with the turbine working at the same rotating speed as at the desired operational conditions. This was done to be able to take the baseline of the gravitational+centrifugal contribution to the load dynamics across the blade instrumented during the wind-on tests, allowing us to separate the aerodynamic driven blade dynamics. The Rayleigh backscattering profiles
were processed with Luna's ODiSI processing toolkit converting reflected light, to strain data. A spatial smoothing filter using





| $\lambda$ | 1 | 2 | 3 | 3.5 | 4 | 5 | 6.5 |
|---|---|---|---|---|---|---|---|
| $\Omega$[rpm] | 56 | 110 | 170 | 195 | 225 | 280 | 365 |
| $N_{acq}^{ROT} = \dfrac{f_{acq}^{\mathrm{RBS}}}{\Omega/60}$ | 107.1 | 54.5 | 35.3 | 30.8 | 26.7 | 21.4 | 16.4 |

**Table 1.** Number of acquisition of strain fields per rotation, for each tip speed ratio assessed in the experimental campaign.

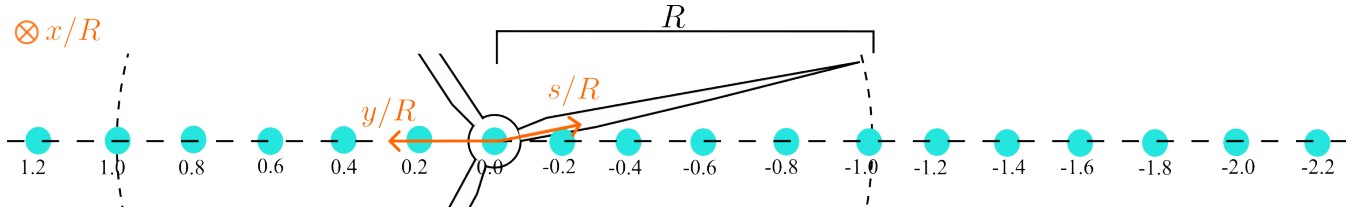

**Figure 3.** Schematic of the hot-wire array set-up locations, along the spanwise direction of the turbine. The wind turbine is mounted at the centre of the test section of the wind tunnel, and the six hot-wire probe array is mounted on a traverse system that moves along the spanwise, and streamwise direction of the wind tunnel allowing the characterisation of the generated wake dynamics by the wind turbine.

a $3^{\mathrm{rd}}$ order Savitzky-Golay filter with a window size of 5 elements was used afterwards to smooth the signal spatially, while preserving the different dynamic spatial characteristics of the blade's response (de Oliveira et al., 2024).

### 2.3 Constant Temperature Anemometry Measurements

The wind turbine's wake dynamics were captured concurrently with the RBS measurements with an array of 6 single velocity-
component hot-wires, positioned downstream of the wind turbine facing the inflow (see the experimental schematic represented in fig. 3). Hot-wire probes were connected to a Dantec Streamline Pro system acquiring at a sampling frequency of $10,000$Hz, ensuring a temporal resolution adequate to resolve the wind turbine wake statistics of interest. The air temperature, and the static pressure of the flow were simultaneously acquired with a temperature probe and a static pressure probe connected to a Furness control system.

The hot-wire probe array was mounted on a traverse spanning for $7$m along the streamwise direction of the turbine. Each probe was calibrated before and after experiments using a standard velocity ramp procedure with concurrent temperature and pressure logging. Probe alignment was verified before each experimental run to ensure data quality and repeatability. To cross-correlate the acquired velocities with the RBS strain signal, the data was filtered with a low-pass filter, and downsampled to match the $100$Hz RBS sampling frequency. The downsampling used did not result in a change of wake statistics presented
in this manuscript. The hot wire probes were set to capture wake dynamics of the turbine, along the spanwise direction from $y/R \in \{-2.2, 1.2\}$, by moving the 6-probe rack along the traverse, encompassing $18$ spanwise measurement locations. The traverse was moved along the streamwise direction over 6 measurement stations: $x/R \in \{1.4, 2, 3, 4, 6, 8\}$, providing detailed measurements in the near-wake of the turbine, often described to start immediately downstream of the turbine, spanning up to



a region between $4 \leq x/R \leq 8$ (Vermeer et al., 2003; Medici and Alfredsson, 2006). Both the RBS and the hot wire rack were
triggered for a total of $T = 120$s. For each acquisition, the hot wire signal was corrected based on the respective experienced
averaged temperature following the procedure of Hultmark and Smits (2010).

The concurrent acquisition of hot-wire and RBS data was achieved through synchronization between Luna's ODiSI-B and
Dantec Streamline Pro system. This was done via hardware triggering, through a shared TTL pulse generator controlled by
an in-house LabView script, adapted from the procedure used in de Oliveira et al. (2024). The assessment of the concurrent
fluctuating strain measurements from the RBS, and the fluctuating velocity fields from the hot-wire measurements, allow us
to link the different coherent flow structures generated in the wake of the wind turbine, and the respective blade's dynamic
response.

## 3 Blade loads in quiescent atmosphere

The loads experienced by a wind turbine blade result from the combined action of gravitational, centrifugal, and aerodynamic
forces (Hansen, 2015; Wen et al., 2020; Jenkins et al., 2021). Assuming that their individual contributions to the blade defor-
mation can be separated into three components, the measured strain ($\varepsilon$) can be decomposed into:

$$\varepsilon = \varepsilon_g + \varepsilon_c + \varepsilon_a, \tag{3}$$

where each of the terms correspond to the gravitational, centrifugal and aerodynamic strain-driven components. The first two
depend mainly on the blade geometry, mass distribution, and material mechanical properties. Instantaneous gravitational loads
are influenced by the phase angle of the blade, and centrifugal loads by the rotating speed of the turbine. The aerodynamic
component includes lift-and-drag related strain, expected to be affected by FST. The effect of the latter is twofold: the presence
of FST imposes a fluctuating velocity field adjacent to the blade, directly influencing the experienced structural dynamics; and
modifies the shedding mechanisms in the wake of the turbine, indirectly contributing to the attained structural response of the
blade. It can be noted that $\varepsilon_a$ may be affected by the Himmelskamp effect, whereby changes in rotor angular velocity alter
the aerofoil's lift and drag coefficients. However, its influence in $\varepsilon_a$ is expected to be minor compared with that arising from
variations in $\lambda$.

To assess the impact of gravitational and centrifugal loads on the measured strain signal, preliminary experiments were
conducted in a quiescent atmosphere ($U_\infty = 0$), with the turbine operating at the matched rotational speed to $\lambda \in [1, 6.5]$. In
addition, an extra test denoted by $\lambda = 0$ was conducted to characterise and assess the impact of $\varepsilon_g$, where the instrumented
blade was fixed at 12 phase angles, and strain was recorded at each of the locations allowing us to remove the $\varepsilon_c$ component
from $\varepsilon^{U_\infty=0} = \varepsilon_g + \varepsilon_c$. Figure 4 *a)* presents the ensemble of the phase averaged strain ($\tilde{\cdot}$) across the blade ($\langle \widetilde{\varepsilon_g + \varepsilon_c} \rangle|_{s/R}$) for
the different rotating velocities tested, and $\lambda = 0$ referring to the case where $\varepsilon_g$ is isolated. This procedure isolates the $\varepsilon_g$ and
$\varepsilon_c$ contributions from total strain measurements. The phase angle of the blade (here denoted by $\phi$) is set so that at $\phi = \pi$ the
instrumented blade points downwards, *i.e.*, in the same direction as the gravity vector.

At $\lambda = 0$, the strain response follows a sinusoidal profile peaking at $\phi = \pi$ where the instrumented blade points downwards
and experiences the largest tensile strain. At $\phi = 0$ the blade points upwards and experiences compressive strain, hence, the





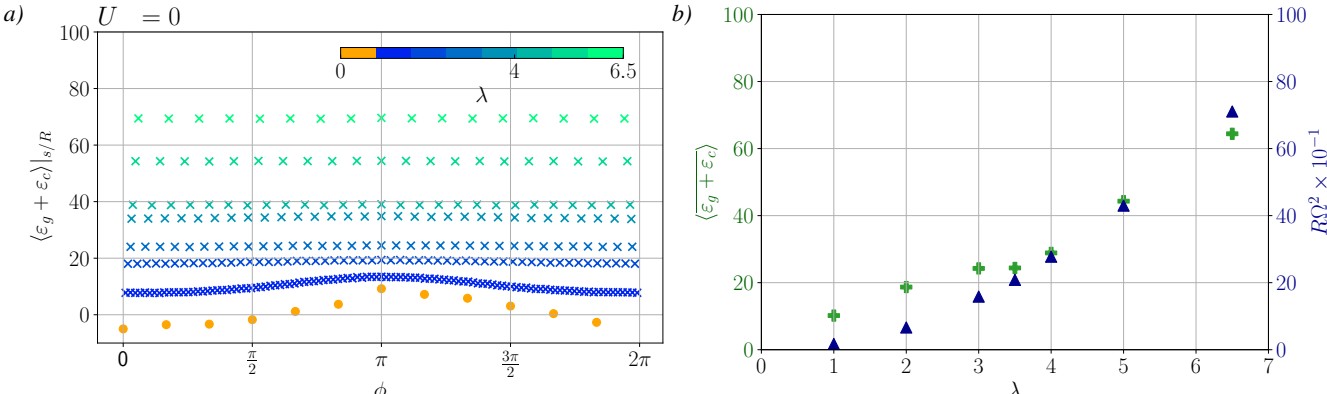

**Figure 4.** *a)*: Phase averaged strain measured across the blade under a quiescent background, averaged over $s/R = [0.15, 0.95]$, at different phases $\phi$, for each of the rotational conditions respective of the tested operating tip speed ratios. *b)*: Ensemble average of strain measurement across the blade's span, for the different tested rotating velocities and expected acceleration induced by centrifugal force $R\Omega^2$.

negative values. As $\lambda$ increases, both the magnitude and the symmetry of the strain variation increase. At $\lambda = 1$, the phase-averaged strain exhibits similar asymmetry to $\lambda = 0$, consistent with a stronger contribution from gravitational loads. As $\lambda$ increases, the strain amplitude increases, highlighting the increased contribution of the centrifugal load to the total strain. In addition, it is observed that the total strain is no longer phase-dependent. At high tip-speed ratios, the corresponding Froude number is large ($Fr(\lambda = 6.5) = \Omega/(\sqrt{(g/R)} \approx 9)$), indicating that the blade loading and resulting strain are predominantly governed by centrifugal forces, whereas the contribution of gravitational loads becomes negligible, potentially due to increased blade stiffening (Meng et al., 2022; Hoskoti et al., 2023).

To validate the measurements, figure 4 *b)* juxtaposes the ensemble average of the acquired strain measured under quiescent-flow conditions, $\langle \varepsilon_g + \varepsilon_c \rangle$, with the classical centrifugal-force scaling $R\Omega^2$, where $R$ corresponds to the blade radius and $\Omega$ to the turbine's angular velocity for each of the tested tip speed ratios $\lambda$. As seen from figure 4 *b)*, $\langle \varepsilon_g + \varepsilon_c \rangle$ scales fairly well with $R\Omega^2$, emphasizing the expected centrifugal scaling, while deviations may arise from aerodynamic loads associated with the turbine operating as a fan under quiescent background conditions, or to a general blade stiffening at increased rotational speeds.

## 4 Impact of $\lambda$ and FST on turbine operation and time-averaged blade loads

The rotor's power coefficient ($C_P$) retrieved from monitoring the generator output under different operating conditions, is one of the key parameters wind turbine comissioners monitor. This coefficient quantifies the efficiency of converting incoming kinetic energy into electrical power. It is strongly dependent on the tip-speed ratio ($\lambda$), which governs the aerodynamic regime experienced by the rotor. The power coefficient ($C_P$) is then defined as:

$$C_P = \frac{P_{\text{motor}}}{1/2 \times \rho_{air}^{20\circ C} \pi R^2 U_\infty^3},$$

(4)



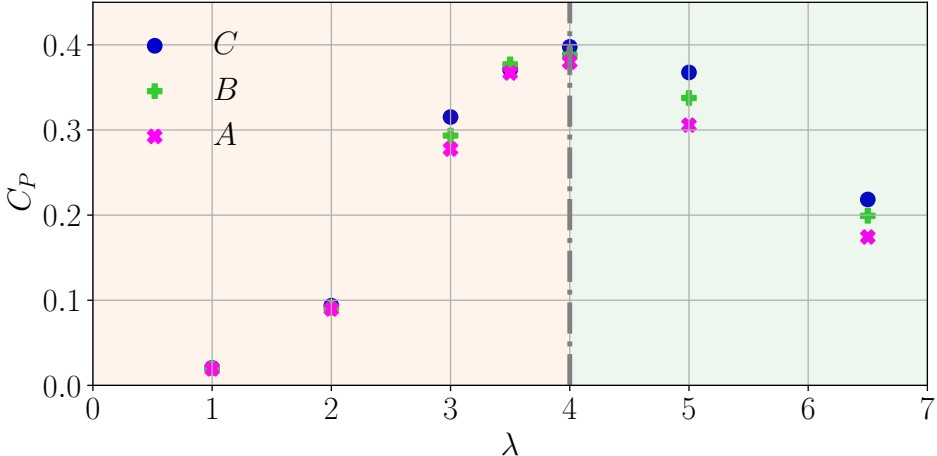

**Figure 5.** Power coefficient ($C_P$) of the wind turbine model as a function of tip speed ratio ($\lambda$).

where $P_{\mathrm{motor}} = VI$ is computed from the measured current and voltage at the generator's terminals during the turbine's operation.

Figure 5 shows the wind turbine's $C_P$ for the different FST and operating conditions tested. As expected, $\lambda = 4$ presents a clear maximum of $C_P$ for the 3 tested FST conditions, where the turbine is operated at design conditions. Beyond this point

($\lambda > \lambda_d$), $C_P$ declines with the increase of $\lambda$ due to a reduction in angle of attack and associated lift. At $\lambda < \lambda_d$, $C_P$ increases with $\lambda$ as the blade moves from stalled conditions to optimum operating conditions. Similarly to Medici and Alfredsson (2006); Tobin et al. (2015); Chamorro et al. (2015); Gambuzza and Ganapathisubramani (2021), we observe increased $C_P$ as $TI$ in the free-stream increases, potentially attributed to an increased lift-drag coefficient (Maldonado et al., 2015; Thompson et al., 2023). Due to wind velocity fluctuations and atmospheric dynamics increasing the variability of the free-stream velocity,

turbines are exposed to a varying range of $\lambda$, and the impact of below-and-above design $\lambda$ (regimes of different aerodynamic turbine performance) on the blade's structural dynamics calls for attention.

The operating condition governs the blade dynamics (Hansen, 2015) and controls the onset and progression of stall across the blade's span. At the same time, modifications of the flow features within the wake feed back into the structural response of the blades, as the inherent wake dynamics are imprinted on them (de Oliveira et al., 2025). This highlights the necessity of

assessing the structural response of a wind turbine's blade, concurrently to the flow dynamics to which the blade is exposed jointly. We start by analysing the time-averaged velocity profiles in the wake of the turbine, under the conditions tested. Figure 6 *a)* shows the contours of velocity deficit, and figure 6 *b)* presents the turbulence intensity ($TI$) profiles for different $\lambda$ values under inflow cases A and C.

The velocity deficit grows with increasing $\lambda$, reflecting the enhanced extraction of momentum from the flow. The presence of

free-stream turbulence increases the complexity of wake development, altering the dynamics and coherence of wake structures (Biswas and Buxton, 2025; Bourhis et al., 2025). The $TI$ profiles reveal a strong sensitivity to inflow turbulence conditions.



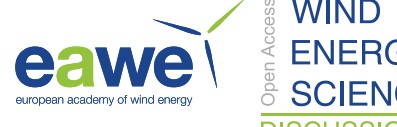

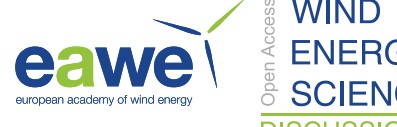

**Figure 6.** *a)*: Velocity deficit across $y/R$, for FST cases $A, C$ and $\lambda \in \{2, 4, 6.5\}$, *b)*: Turbulence intensity profiles across $y/R$ for FST cases $A, C$ and $\lambda \in \{2, 4, 6.5\}$. The dotted points refer to the spatial positions in which hot-wires were positioned.



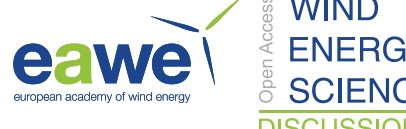

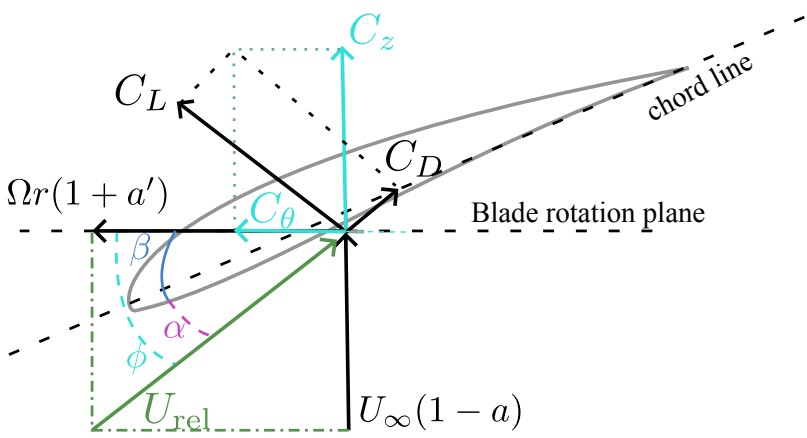

**Figure 7.** Velocity triangle schematic highlighting the induced velocities for a section of the blade.

For $\lambda > \lambda_d$, the $TI$ within the shear layer separating the wake and the free-stream increases markedly, coinciding with an increased thrust coefficient and increased momentum deficit across the turbine's wake. As a result, the blade experiences a range of distinct flow-induced dynamics across its span, from the blade's root, evolving towards the tip.

The impact of operating conditions of the turbine and FST on the time-averaged loads to which the turbine's blade is exposed is assessed by taking the time-average ($\bar{\cdot}$) of the aerodynamic strain $\overline{\varepsilon_a}$ computed from:

$$\overline{\varepsilon_a}(s/R) = \overline{\varepsilon}(s/R) - \overline{\varepsilon_{g+c}}(s/R), \tag{5}$$

serving as a surrogate for the time-averaged distribution of the experienced aerodynamic loads. Before delving into interpretation of the results, it is relevant to assess the driving aerodynamic loading mechanisms across the turbine's blade. Figure 7

presents a schematic of the velocity triangle on an aerofoil profile, where $\beta$ corresponds to the aerofoil's pitch angle, $\alpha$ to the local angle of attack, $\phi$ to the flow angle with respect to the aerofoil, $a$ to the axial induction factor and $a'$ to the tangential induction factor. Moreover, $U_{\text{rel}}$ refers to the relative velocity vector and $r$ to the radial distance of the profile to the centre of the rotor (Hansen, 2015; Bourhis et al., 2022).

In figure 7, $C_D$, $C_L$ correspond respectively to the drag, lift coefficients, and $C_z$ and $C_\theta$ to the projected lift and drag

coefficient components into the rotor plane, computed as:

$$\Rightarrow \begin{cases} C_z = C_L \cos(\phi) + C_D \sin(\phi), \\ C_\theta = C_L \sin(\phi) - C_D \cos(\phi). \end{cases} \tag{6}$$

The elementary torque ($d\tau$) and thrust ($dT$) acting on the aerofoil section can be computed as:

$$\Rightarrow \begin{cases} dT = \frac{1}{2} \rho c U_{\text{rel}}^2 C_z r \, dr, \\ d\tau = \boldsymbol{r} \times d\mathbf{F_\theta} = \frac{1}{2} \rho c U_{\text{rel}}^2 C_\theta r^2 \, dr, \end{cases} \tag{7}$$

where $c$ corresponds to the local blade chord length.





The decomposed strain measurements along the flapwise ($\varepsilon^f$) and edgewise ($\varepsilon^e$) components are driven by induced bending moments across the blade's profile, and are related to $dT$ and $d\tau$:

$$\varepsilon^f \approx \frac{M^f}{EI^f} \approx \frac{\int r dT}{EI^f}, \qquad \varepsilon^e \approx \frac{M^e}{EI^e} \approx \frac{\int d\tau}{EI^e}, \tag{8}$$

where $E$ refers to the material's Young's modulus, and $I^f$ and $I^e$ to the flapwise and edgewise moment of inertia. Given that for our blade $I^f < I^e$, the aerodynamic-induced flapwise strain is expected to be significantly larger than its edgewise counterpart:


$$I^f < I^e \rightarrow \varepsilon^f > \varepsilon^e. \tag{9}$$

    Consequently, the edgewise strain component ($\varepsilon^e$) exhibits much lower magnitudes and is more susceptible to measurement uncertainty. In addition, due to the geometry of the blade, the flapwise bending moment also induces a deformation along the edgewise direction. This coupling effectively contaminates the measured $\varepsilon^e$ field, particularly where the fibre's sensing axis is

not aligned with the local chord. To minimise this cross-sensitivity, only locations where the fibre orientation closely follows the chordwise direction are considered in the analysis of the edgewise aerodynamic strain component.

    Figure 8 presents the distributions of $\overline{\varepsilon_a^f}$ and $\overline{\varepsilon_a^e}$ for FST cases A, C, and 3 representative tip-speed ratios ($\lambda \in \{2, 4, 6.5\}$), reflecting below-design, design and above-design operating conditions. The time-averaged flapwise ($\overline{\varepsilon_a^f}$) and edgewise ($\overline{\varepsilon_a^f}$) aerodynamic induced strain components are decomposed from the time-averaged aerodynamic induced strain ($\overline{\varepsilon_a}$) as:

$$\overline{\varepsilon_a^f} = \overline{\varepsilon_a}\cos(\theta_f(s/R)), \qquad \overline{\varepsilon_a^e} = \overline{\varepsilon_a}\sin(\theta_f(s/R)) \tag{10}$$

where $\theta_f(s/R)$ corresponds to the local angle of the fibre path across the blade's spanwise extent.

    At $\lambda = 2$ and FST case A, the blade operates in a stalled regime, where flow separation and high angles of attack result in poor aerodynamic efficiency of the entire blade's span. At these conditions, the distributions of $\overline{\varepsilon_a^f}$ are minimal and broadly distributed. At $\lambda = 2$, the axial thrust force acting across the rotor's plane is expected to be smaller than at larger $\lambda$, resulting in a

decreased flapwise deformation of the blade. This is observed by the smaller magnitude of the profiles of $\overline{\varepsilon_a^f}$ at $\lambda = 2$ relatively to $\lambda \in \{4, 6.5\}$, as presented in figure 8. Moreover, under these conditions ($\lambda = 2$), the blade's tip region shows a negligible magnitude of $\overline{\varepsilon_a^f}$ resulting from smaller experienced bending stresses in this section of the blade. As $\lambda$ increases up to the design point ($\lambda = 4$), the local angle of attack across the blade approaches the optimum angle of attack of the blade's aerofoils, and the rotor operates under optimal aerodynamic efficiency. The distribution of $\overline{\varepsilon_a^f}$ becomes more structured, with clearer

peaks (with a respective increase in magnitude as well) across the span of the blade, reflecting the increased flapwise bending stresses introduced by larger thrust acting on the blade. At above-design conditions ($\lambda = 6.5$), the blade operates at low angles of attack, with a reduction of generated lift across the blade—particularly near the tip. However, due to a subsequent increase of the axial thrust acting on the rotor, $\overline{\varepsilon_a^f}$ increases in magnitude across the blade relatively to the conditions experienced at $\lambda = 4$.

For FST case C, the profiles of $\overline{\varepsilon_a^f}$ increase in magnitude compared of FST case A, potentially linked an increased lift coefficient under turbulent free-stream conditions (Maldonado et al., 2015; Thompson et al., 2023). This increase is especially





**Figure 8.** Spanwise time-averaged aerodynamic strain contribution to: *a)* flapwise ($\overline{\varepsilon_a^f}$); and *b)* edgewise ($\overline{\varepsilon_a^e}$) strain components, for representative tip-speed ratios. Locations where the fibre optic's axis is not aligned with the chordwise axis of the blade's profile are shaded in the distribution of $\overline{\varepsilon_a^e}$.



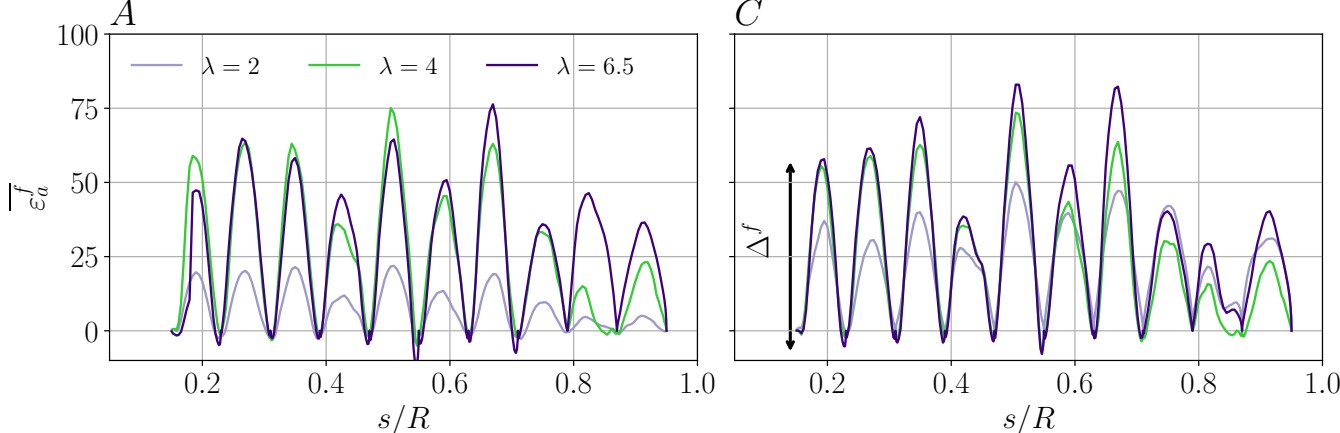

**Figure 9.** Time-averaged flapwise strain along the spanwise extent of the blade for 3 different $\lambda$, and FST cases $A$ and $C$.

observed for $\lambda = 2$, suggesting that under $\lambda < \lambda_d$, the blades are more susceptible to FST. This is potentially linked to reattachment of the flow on the suction side of the blade, enhanced by the presence of FST. Furthermore, the midspan-to-tip region of the blade is the section that most experiences this increase in flapwise stresses introduced by the presence of increased $TI$ in the FST. These results underscore the strong coupling between operating condition and inflow turbulence to the time-averaged structural response.

The distributions of $\overline{\varepsilon_a^e}$ show a less distinct and noisier evolution along the span, consistent with the lower measurement sensitivity. Negative values of $\overline{\varepsilon_a^e}$ present regions where the blade experienced compressive stresses with respect to the no-wind centrifugal+gravitational baseline. Measurements at $\lambda = 2$ present the most pronounced variations with increased $TI$ in the inflow, with the tip region transitioning from compressive, to tensile behaviour, when neglecting centrifugal+gravitational effects from the flow and operation induced strain. For $\lambda > \lambda_d$, the influence of the operating conditions becomes less substantial, marked by slight increases of the local magnitude of $\overline{\varepsilon_a^e}$ across the blade's span. Overall, the comparatively weaker and less organized patterns of $\overline{\varepsilon_a^e}$ reinforces that the blade's structural response is primarily governed by flapwise dynamics, related to modifications in the wind turbine's thrust coefficient ($C_T$).

Figure 9 presents the distributions of $\overline{\varepsilon_a^f}$ across $s/R$, for $\lambda \in \{2, 4, 6.5\}$ and FST cases A and C. The signal is characterised by a sequence of peaks and troughs. Each peak across the distribution of $\overline{\varepsilon_a^f}$ corresponds to a point along the fibre sensing path where the fibre axis is aligned with the blade's spanwise vector, where the sensors experience primarily flapwise stresses. Each trough to a point where the fibre axis is aligned with the chordwise extent. As $\lambda$ increases, so does the overall experienced magnitude of $\overline{\varepsilon_a^f}$ across the blade in-line with the increase in the thrust coefficient of a wind turbine with increasing $\lambda$. Higher free-stream $TI$ increases the magnitude of the peaks of the distribution of $\overline{\varepsilon_a^f}$, reflecting an increased flapwise bending moment. $\Delta^f$ allows us to quantify the induced aerodynamic bending on the blade's sections along it's extent. Together, figures 8 and 9 show that $\lambda$ governs both the magnitude and spatial organisation of the mean aerodynamic strain, while FST modulates its amplitude.





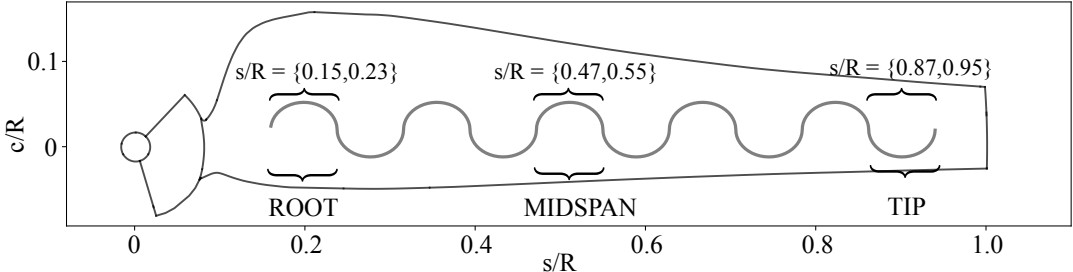

**Figure 10.** Nomenclature of blade sections and integration regions, characterising the regions of the blade of most interest. The ROOT consists of the region where most stresses accumulate, the TIP is where tip vortices, the strongest coherent flow structure present in the immediate near wake of the turbine, are the most intense. The MIDSPAN comprises a region of transition from ROOT to TIP dynamics.

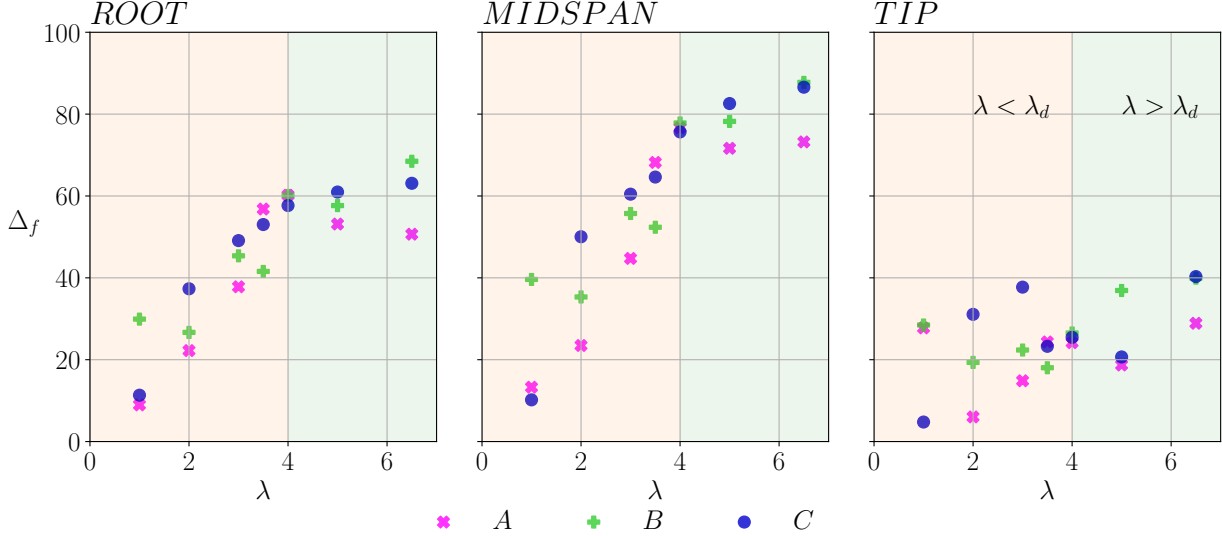

**Figure 11.** Evolution of $\Delta_f$ as the flapwise bending proxy in the blade's ROOT, MIDSPAN and TIP.

To assess the effects of FST and $\lambda$ on the different sections of the blade, we now divide it into three regions of interest
350 denoted by ROOT, MIDSPAN, and TIP, as illustrated in fig. 10. These segments are used consistently throughout this work.

The evolution of the magnitude of the bending proxy $\Delta_f$ as a function of $\lambda$ and FST (presented in figure 11), obtained from the peak-trough strain differences within each section, reveal distinct trends strongly influenced by $\lambda$ and the section of the blade under analysis. $\Delta_f$ shows a monotonic increase with $\lambda$ across all spanwise locations. At the ROOT, the growth is gradual with $\lambda$ until reaching $\lambda_d$. At above-design conditions, $\Delta_f$ increases but at a slower rate. The MIDSPAN shows
355 a similar behaviour, but with a steeper increase of $\Delta_f$ both at below and above the design operating conditions. The abrupt change marking the transition from below-to-above design operating conditions is still observed. The TIP also presents an





increase of $\Delta_f$, but at a slower rate than the other sections under analysis. The decreased rate of increase for $\lambda > \lambda_d$ reflects the aerodynamic stabilisation of the blade. Moreover, $\Delta_f$ follows closely the distribution of the thrust coefficient as a function of $\lambda$ as described in Medici and Alfredsson (2006), where the turbine's thrust coefficient increased with $\lambda$ up to $\lambda_d$, with a slower increase at $\lambda > \lambda_d$, reflecting the link between the axial thrust acting on the rotor, with the evolution of the measured $\Delta_f$ in this work.

Increased $TI$ in the FST amplifies the overall experienced $\Delta_f$, particularly for $\lambda > 1$ suggesting an increased thrust coefficient. Moreover, at design operating conditions, the variation of $\Delta_f$ introduced by the different FST cases tested is the smallest at all 3 blade sections analysed. This suggests that at design conditions, the effects of FST on the time-averaged loads are mitigated by the operational conditions of the turbine. Furthermore, figure 11 demonstrates a reduced sensitivity of $\Delta_f$ to FST at optimum tip speed ratio.

## 5 Fluctuating blade dynamics across operating conditions and FST "flavours"

Having assessed the blade's time-averaged strain, we now assess the impact of $\lambda$ and FST on fluctuating induced strain ($\varepsilon' = \varepsilon - \bar{\varepsilon}$) dynamics across the blade's span. The turbine's rotational speed fluctuations reached a maximum of $\Delta\Omega = 9$rpm at $\lambda = \lambda_d$. Even though the variations of the rotational velocity $\Omega$ are relatively small ($\Delta\Omega/\Omega \approx 4\%$ at $\lambda_d$), their quadratic influence on the centrifugal load ($F_c \propto mr\Omega^2$) makes their contribution non-negligible to the total strain fluctuations ($\mathrm{d}F_c \propto 2mr\Omega\mathrm{d}\Omega$).

Hence, while the absolute magnitude of $\Delta\Omega$ is small, the fluctuating centrifugal load introduces a component $2\Omega\Delta\Omega$ that can significantly influence the fluctuating strain. This term must be therefore accounted for when interpreting $\mathrm{RMS}(\varepsilon')$ in terms of aerodynamic versus inertial contributions. In addtion, the sensitivity to the gravitational induced strain decreases with $\lambda$ as previously seen, as a product of blade stiffening. To account for these effects, we removed $\mathrm{RMS}(\varepsilon'_{g+c})$ acquired in quiescent atmosphere (*cf.* preliminary experiments described in section 3) to the $\mathrm{RMS}(\varepsilon')$ acquired with the wind on, allowing us to estimate the aerodynamic-induced fluctuating strain component $\mathrm{RMS}(\varepsilon'_a)$ from:

$$\mathrm{RMS}(\varepsilon'_a) = \mathrm{RMS}(\varepsilon') - \mathrm{RMS}(\varepsilon'_{g+c}), \tag{11}$$

assuming that the aerodynamic-induced fluctuating strain is uncorrelated to the experienced gravitational+centrifugal induced fluctuating strain across the blade. Figure 12 presents the aerodynamic contribution to strain fluctuations ($\mathrm{RMS}(\varepsilon'_a)$) at the different sections of the blade for the different test cases.

Two robust features emerge. Firstly, the TIP consistently exhibits the largest fluctuation levels across all operating conditions and FST cases. This points to the strong imprint of tip-vortex shedding and broadband perturbations, which dominate the unsteady aerodynamic environment in this region; and to the impact of having a region of thinner aerofoil profiles reflecting on a smaller local moment of inertia. Secondly, a marked increase in overall strain fluctuations at $\lambda \approx 3.5$ is observed, potentially emphasizing the influence of the unstable regime of partially stalled to partially attached flow conditions. This local peak highlights the strong coupling between stall dynamics and the blade's-tip induced fluctuations.

The aerodynamic-regime dependence of the blade fluctuating dynamics is further clarified by assessing the impact of $\lambda$ on $\mathrm{RMS}(\varepsilon'_a)$. At $\lambda < \lambda_d$, $\mathrm{RMS}(\varepsilon'_a)$ increases with $\lambda$ from below-to near-design conditions. At $\lambda \geq \lambda_d$, $\mathrm{RMS}(\varepsilon'_a)$ decreases in all

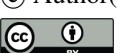



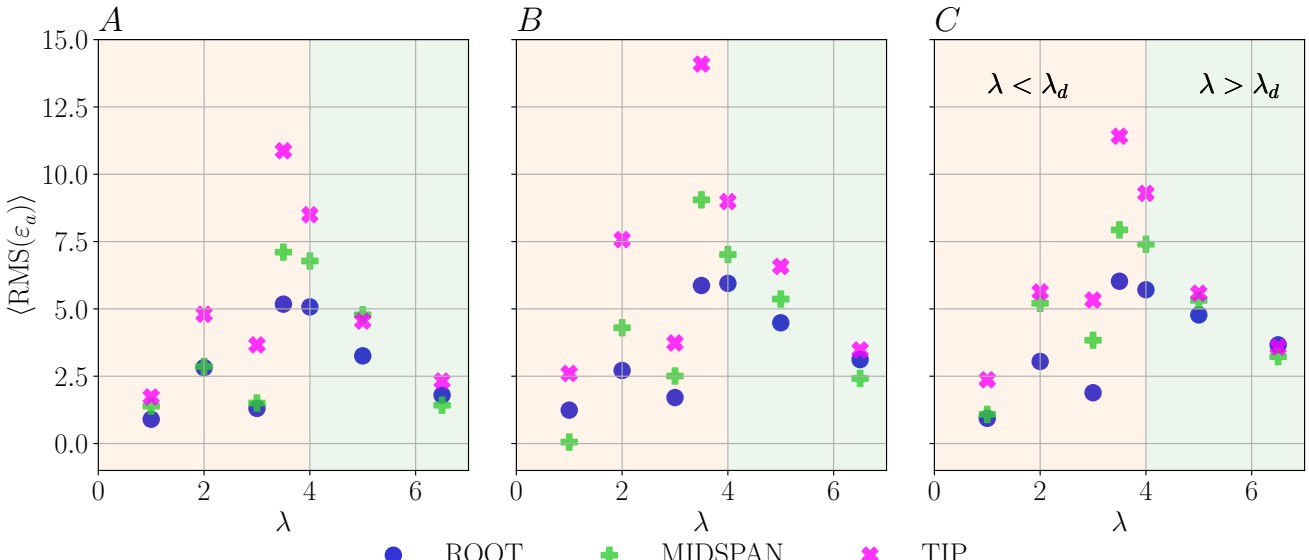

**Figure 12.** Root-mean-square strain $\mathrm{RMS}(\varepsilon_a')$ across blade regions, for the operating conditions and FST cases tested.

3 sections of the blade, closely following the $C_P$ curve's behaviour in this range of $\lambda$ hinting at the link between $\mathrm{RMS}(\varepsilon_a')$ and the wind turbine's performance. Furthermore, even though at $\lambda < \lambda_d$ portions of the blade operate under stalled conditions, we observe that $\mathrm{RMS}(\varepsilon_a')$ presents similar values to $\mathrm{RMS}(\varepsilon_a')$ at $\lambda \geq \lambda_d$, where the flow is mostly attached.

The influence of FST is secondary: FST case C produces a mild amplification of fluctuations compared to FST case A. FST case B presents the largest values of $\mathrm{RMS}(\varepsilon_a')$, especially at the MIDSPAN and TIP at $\lambda = 3.5$. This is potentially due to an additional effect of the integral length scale of FST being of the order of the chord length of the blade at the MIDSPAN region (Maldonado et al., 2015). However, the influence of FST remains relatively smaller than the systematic dependence on aerodynamic performance of the blade ($\lambda$). Moreover, at $\lambda \geq \lambda_d$, conditions in which the flow is attached to the blade, increased $TI$ consistently increases $\mathrm{RMS}(\varepsilon_a')$ across the three sections of the blade.

Taken together, these observations show how TIP dynamics persist as the dominant source of fluctuating loads in the blade, potentially driven by both the presence of tip vortices generated at this section of the blade (Yang et al., 2012; Biswas and Buxton, 2024b; Bourhis et al., 2025), combined with a thinner profile with smaller inertia, reacting stronger to induced dynamics. The proximity of maximum $\mathrm{RMS}(\varepsilon')$ to $\lambda_d$ highlights the trade-off between aerodynamic efficiency and structural excitation. These results suggest that, from an aerodynamic-induced fatigue damage perspective, it is preferable to maintain wind turbines operating at slightly above design conditions with the compromise of the increased contribution of centrifugal-loads, rather than slightly-below design.

Figure 13 *a)* presents the ensemble average of $\mathrm{RMS}(\varepsilon_{g+c}')$, and $\mathrm{RMS}(\varepsilon')$ across the blade's span ($\langle\cdot\rangle|_{s/R}$), as a function of $\lambda$ and FST.







**Figure 13.** *a)*: Ensemble average across the blade of the RMS of the gravity-induced strain fluctuations ($\varepsilon'_{g+c}$), and aerodynamic-induced strain fluctuations ($\varepsilon'_a$) as a function of FST and $\lambda$. *b)* ratio between RMS($\varepsilon'_a$) and RMS($\varepsilon'$) ($\zeta = \mathrm{RMS}(\varepsilon'_a)/\mathrm{RMS}(\varepsilon')$) at the ROOT, MIDSPAN and TIP, as a function of $\lambda$ and FST conditions.





The influence of $\mathrm{RMS}(\varepsilon_g')$ is expected to be constant across all $\lambda$. The variations across $\lambda$ of $\mathrm{RMS}(\varepsilon_{g+c}')$ are introduced by fluctuations of the rotational speed as discussed in section 3. By contrast, $\mathrm{RMS}(\varepsilon_a')$ shows a strong dependence on $\lambda$ and FST

conditions.

The relevance of the contribution of aerodynamic effects is quantified by the ratio $\zeta = \frac{\mathrm{RMS}(\varepsilon_a')}{\mathrm{RMS}(\varepsilon')}$. Figure 13 *b)* presents the distribution of $\zeta$ at the ROOT, MIDSPAN and TIP, for the different $\lambda$ and FST conditions. At the ROOT, $\zeta$ presents the smallest magnitude across all conditions, confirming that strain fluctuations in this region are mainly dominated by gravitational effects. The TIP presents the largest magnitude of $\zeta$ across all operational regimes, highlighting the role of aerodynamic loads

in this particular region of the blade. The MIDSPAN acts once more as a transitional region between TIP to ROOT dynamics. Moreover, these results show the relevance of aerodynamic induced stresses on the blade especially at the TIP, where the aerodynamic contributing to the strain fluctuations reaches close to $\approx 75\%$ of the total strain fluctuations when the turbine operates at $\lambda \approx \lambda_d$. At $\lambda > \lambda_d$ the aerodynamic contribution decreases compared with $\zeta$ at $\lambda = \lambda_d$. This is due to a lower magnitude lift force acting on the aerofoil profiles across the blade, driven by a reduced local angle of attack across the blade's

span.

Figure 15 presents the probability density functions (PDFs) of standard-deviation-normalised strain fluctuations ($\varepsilon^{\star}$), at different blade locations computed as:

$$\mathrm{PDF}_{\varepsilon'} = \mathrm{PDF}(\sum_{s_i}^{s^e} \varepsilon^{\star}(s/R)), \tag{12}$$

where $s_i$ and $s_e$ correspond to the start and end coordinates of each respective region ROOT/MIDSPAN/TIP under anal-

ysis. The PDFs are presented for FST conditions A and C, and quiescent background conditions to infer the impact of the aerodynamic events acting on the blade on the distribution of fluctuating dynamics.

The probability density functions (PDFs) of the standard-deviation-normalised strain show broadly similar shapes across operating conditions and blade locations. The PDFs under quiescent background conditions follow a Gaussian-like profile, consistent with the periodic impact of the combined centrifugal+gravitational loads acting on the blades. At below-design

operation ($\lambda = 2$), the distributions display mild asymmetry, particularly at the ROOT and MIDSPAN. With increasing $\lambda$, they become more symmetric and exhibit slightly heavier tails, suggesting a weak increase in the likelihood of extreme fluctuations under more periodic excitation. Spanwise differences are modest: the ROOT remains the most Gaussian-like, the MIDSPAN shows intermediate behaviour, and the TIP displays minor tail broadening, likely associated with tip-vortex dynamics and broadband perturbations. The influence of free-stream turbulence is secondary to that of $\lambda$ and spanwise position, manifesting

mainly as subtle changes in tail amplitude. Nevertheless, variations in FST influence $\mathrm{RMS}(\varepsilon')$ (as seen in figure 18), such that while the normalised distributions remain largely self-similar, the absolute magnitude of strain fluctuations is affected by the presence of FST.

We can now assess how the velocity fluctuations in the wake of the turbine map to strain fluctuations across the blade's span. Figure 15 presents the PDFs of velocity fluctuations across the span of the turbine at $x/R = 1.4$ and $x/R = 8.0$, for

$\lambda \in \{2, 4, 6.5\}$ and FST cases A and C.



**Figure 14.** PDFs of standard-deviation-normalised strain fluctuations for representative $\lambda$, FST conditions, quiescent background conditions $(NW)$, and blade locations.

For all $\lambda$, case C produces broader, flatter distributions than case A, reflecting the impact of enhanced velocity fluctuations present in the inflow. The most pronounced differences appear near the tip region of the wake ($|y/R| \approx 1$), where tip-vortex dynamics and shear-layer perturbations dominate. Closer to the wake centreline, FST broadens the PDF associated with the wake of the nacelle. The organisation of velocity fluctuation PDFs with $\lambda$ mirrors the spanwise strain statistics across the blade: at $\lambda < \lambda_d$, broad velocity PDFs reflect disordered stalled flow, consistent with the large $\mathrm{RMS}(\varepsilon'_a)$ and heavy-tailed strain distributions at the tip (figures 12, 14). At $\lambda_d$, coherent tip and root vortices strengthen, producing persistent lateral signatures in the wake ($|y/R| \approx 1$ and $|y/R| < 0.4$). At above-design operation, the narrowing of the velocity PDFs around $u'/U_\infty = 0$ reflects reduced wake intermittency and is consistent with the drop in strain fluctuation amplitudes and the lower aerodynamic contribution $\zeta$ across the blade (figure 13).







**Figure 15.** PDFs of velocity fluctuations at $x/R = 1.4$ (*a*)) and $x/R = 8.0$ (*b*)) for representative $\lambda$ and FST cases.



Finally, the downstream evolution from $x/R = 1.4$ to $x/R = 8.0$ shows a progressive narrowing of the PDFs across all $\lambda$ and FST cases, as the wake turbulence undergoes mixing and homogenisation. This attenuation of extreme velocity fluctuations parallels the reduced strain intermittency observed at above-design $\lambda$, confirming the link between wake homogenisation and smoother blade forcing.

## 6    Spatio-Spectral Dynamics of Blade-Flow coupled system

We now delve into the spectral signature of both the simultaneously acquired strain, and flow fields in the wake of the turbine. This allows us to analyse how the flow dynamics are imprinted onto the blade dynamics. Figure 16 presents the power spectral densities (PSDs) of the turbine's wake velocity fluctuations ($E(u'/U_\infty)$) at $x/R = 1.4$ and $x/R = 8$ ($a$) and $b$)), and the PSDs of aerodynamic-induced strain fluctuations ($E(\varepsilon_a')$) along the blade's span (figure 17), for the 3 tested FST cases, and $\lambda \in \{2, 4, 6.5\}$. The profiles of $E(\varepsilon_a')$ are estimated from $E(\varepsilon') - E(\varepsilon_{g+c}')$. The profiles of $E(u'/U_\infty)$ are shown as a function
of $y/R$, presenting the energy content along the wake's spanwise extent. The frequency of the PSDs is normalised as $St_\Omega$, computed as $St_\Omega = f/\Omega^\star$ (where $\Omega^\star = \Omega/60$ provided $\Omega$ is given in [rpm]), such that the peak associated with the blade passing frequency (BPF) is defined by $St_\Omega = 3$.

        Across all conditions, the near-wake velocity fluctuations' spectra (figure 16 $a$)) are dominated by a vertically coherent peak at $St_\Omega = 3$, indicating the 3-bladed rotor's periodic influence being imprinted into the near-wake's velocity field. The
sharpness of this peak, and the level of surrounding broadband energy, depend strongly on operating point. At $\lambda = 2$, the BPF peak is embedded within a high background of broadband fluctuations, driven by the shed vorticity by the turbine's blades in stalled conditions. At $\lambda = 4$, the BPF peak becomes narrower and more distinct, reflecting the emergence of more coherent tip and root vortex shedding. At $\lambda = 6.5$, the wake spectrum is further sharpened around the BPF. The last two operating points present suppressed broadband energy content across most of the turbine's span, indicating a more stable, periodic wake
structure, as a result of decreased flow separation from the blades (Wang et al., 2014; Maldonado et al., 2015). Superimposed on this $\lambda$-dependence, FST modulates the energy across the spectra: larger $TI$ levels elevate the broadband energy content and reduce lateral coherence, especially near the tip region $|y/R| \approx 1$ and root region $0.2 > |y/R| > 0$, similarly to what has been reported by Bourhis et al. (2025). The presence of a peak at the BPF remains unaffected with the introduction of FST, and it remains the dominant dynamic feature observed in the wake, particularly at the tip region of the turbine. In addition to the BPF
peak in the profiles of $E(u'/U_\infty)$, peaks at $St_\Omega \in \{1, 2\}$ become increasingly discernable as $\lambda$ increases. The $St_\Omega = 1$ band is most visible about the tip shear layer region and $|y/R| \approx 1$. $E(u'/U_\infty)$ at $St_\Omega = 2$ presents a smaller magnitude, peaking at the tip shear layer of the turbine's wake, likely emerging as a product of triadic interaction between $St \in \{1, 3\}$ motions (Biswas and Buxton, 2024a). Both these bands are weakened with the presence of FST. Moreover, a peak at $St_\Omega = 1.5$ appears at $\lambda = 4$ around the centreline of the turbine's wake, suggesting the presence of a sub-harmonic of hub vortices appearing
at $|y/R| < 0.2$. At $x/R = 8$ (figure 16 $b$)), all the aforementioned flow structures have mostly dissipated and no clear peaks across the profiles of $E(u'/U_\infty)$ can be observed. Furthermore, at this streamwise location, the spectra are mostly characterised







**Figure 16.** Power-Spectral density (PSD) of the velocity fluctuations in the wake acquired at $x/R = 1.4$ and $x/R = 8$ (panels *a)* and *b)* respectively).





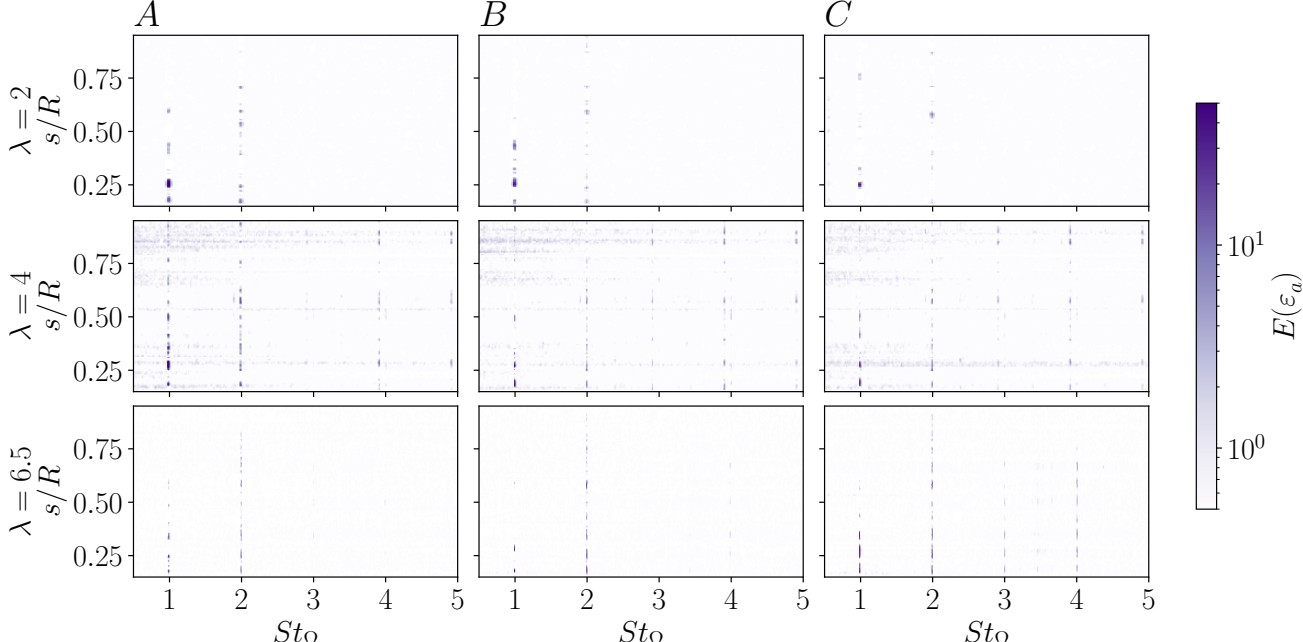

**Figure 17.** PSD of $\varepsilon'_a$ along the blade's span ($s/R$).

by energy at low frequencies. All the tested cases show increased broadband energy content at the root region of the turbine $0.2 > |y/R| > 0$ driven by the nacelle's wake.

The profiles of $E(\varepsilon'_a)$ (figure 17) show a clear influence of $St_\Omega \in \{1,2\}$, selectively appearing at fibre locations that run
parallel to the blade's spanwise vector—supporting the observation made in section 4 that the flapwise bending stresses are dominant. A strong peak at $St_\Omega = 1$ dominates at the root ($s/R < 0.3$) for all cases. Its amplitude decays toward the tip of the blade. The contribution of $St_\Omega = 2$ is relatively constant throughout the blade's span, and shows increased energy across the span at above-design operating conditions ($\lambda \geq \lambda_d$). Moreover, at $\lambda \geq \lambda_d$ clear peaks form along the blade at frequency bands of $St_\Omega \in \{3,4\}$. The energy at the BPF ($St_\Omega = 3$) increases as well, coincident with the increased coherence of tip
vortices at the edge of the blade. The spectra's energy distribution is characterised by a sinusoidal pattern, induced by the layout of sensors used. We observe that sensing points where the fibre is aligned along the spanwise direction present a peak in magnitude, and sensing points where the fibre was aligned with the chordwise direction present a trough. This suggests that the chordwise stress fluctuations are significantly less affected by the wake's coherent dynamics, whilst largely being driven by the aerodynamic performance of the blade, and respective distribution of loads across its span.
At below-design operating conditions, the increasing intensity of the FST diminishes the relevance of coherent flow structures above $St_\Omega = 2$. For $\lambda \geq \lambda_d$, the energy associated with $St_\Omega \in \{1,2,3\}$ seems to increase with the increase of $TI$ in the FST. In addition, it is important to highlight that even though low $\lambda$ generated broadband energy content in the wake of





the turbine—immediately downstream of the central nacelle region—we observe that this doesn't necessarily imprint into the experienced blade dynamics.

Together, these PSDs reveal that the coupled blade-wake system dynamics is anchored to the BPF. The $\varepsilon_a'$ PSDs highlight how the blade acts as a filtering mechanism of flow structures generated in its wake along its flapwise extent, with a diminished effect across its edgewise component, and how different flow structures generated in the wake of the turbine contribute to the different dynamics across the sections of the wind turbine blade.

### 6.1 Integrated Spectral Energy Across Frequencies

To quantify the spatial contribution of the coherent flow structures identified to be relevant in the previous section to the overall dynamics, we integrate the power spectral densities over narrow frequency bands, centered at the most relevant characteristic nondimensional frequencies $St_\Omega \in \{1, 2, 3\}$. The resulting integrated quantities, $\Gamma(\varepsilon')$ and $\Gamma(u')$, represent the local contribution of each rotor frequency harmonic to the total fluctuation energy in strain and velocity fields, respectively, computed as:

$$\Gamma(\Phi)^{St_\Omega} = \int\limits_{r_i}^{r_e} \int\limits_{St_\Omega - W/2}^{St_\Omega + W/2} E(\Phi) \mathrm{d}St_\Omega \mathrm{d}r, \tag{13}$$

where $r_i$ and $r_e$ correspond to the starting and end spatial points over which the integration is computed (for $\Gamma(u')$ $y/R \in [-1.2, 1.2]$ and for $\Gamma(\varepsilon_a')$ $s/R \in [0.15, 0.95]$); $W$ to the window used for the integration over the central frequency $St_\Omega$, set to be $W = 0.2$; and $\Phi$ to the quantity under analysis ($u'$ and $\varepsilon_a'$). $\Gamma(\varepsilon_a')$ is once more estimated by $\Gamma(\varepsilon_a') = \Gamma(\varepsilon') - \Gamma(\varepsilon_g)$, assuming fluctuating contributions from centrifugal sources are negligible, and that $\varepsilon_a'$ and $\varepsilon_g'$ are uncorrelated. Figure 18

presents $\Gamma(u')$ and $\Gamma(\varepsilon_a')$ for the range of $\lambda$ and FST cases tested.

In the wake (figure 18 a)), $St_\Omega = 3$ dominates across all operating points and turbulence intensities, especially at the first streamwise station ($x/R = 1.4$). Its energy decreases with downstream distance, consistent with the progressive breakdown of tip and root vortex coherence along the wake, and decreases with increasing $\lambda$. By contrast, the energy associated with $St_\Omega = 1$ and $St_\Omega = 2$ remains smaller, becoming increasingly energetic as $\lambda$ increases. The energy decay along the streamwise direction,

associated with these structures is not as clear as for $St_\Omega = 3$. However, we can see that for $\lambda = \lambda_d$, $St_\Omega = 1$ increases in energy as the wake evolves along the streamwise direction. This is likely tied to the nonlinear triadic system within the turbine's wake, where energy flows from $St_\Omega = 3$ to $St_\Omega = 1$ in the presence of $St_\Omega = 2$ (Biswas and Buxton, 2024a). Furthermore, the increase in $TI$ in FST tends to increase the energy of $St_\Omega \in \{1, 2\}$, especially in the near-wake and for above-design and design conditions, while decreasing the energy associated with $St_\Omega = 3$ in the near wake when $\lambda \geq \lambda_d$. This is in line with

Biswas and Buxton (2025); Bourhis et al. (2025), where increased $TI$ in the free-stream is associated with a breaking down of coherent tip vortices.

The aerodynamic-induced strain fluctuation (figure 18 b)) dynamics exhibit complementary energy distributions as a function of the location of the blade. $\Gamma(\varepsilon_a')^{St_\Omega = 1}$ dominates across all conditions and spanwise locations especially at the ROOT. The



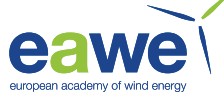
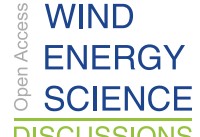

**Figure 18.** $\Gamma(u')$ and $\Gamma(\varepsilon_a')$ at $St_\Omega \in \{1,2,3\}$ across $\lambda$, $x/R$ and FST conditions.





TIP is slightly energised by this flow structure at $\lambda = 3.5$, the onset to design conditions, suggesting higher susceptibility
to such flow structures. The distribution of $\Gamma(\varepsilon'_a)^{St_\Omega=2}$ follows closely $\Gamma(\varepsilon'_a)^{St_\Omega=1}$, with increased relevance at $\lambda < \lambda_d$ at
MIDSPAN. Above design conditions, the MIDSPAN consistently presents the lowest $\Gamma(\varepsilon'_a)^{St_\Omega=2}$ suggesting that both the
ROOT and TIP are the most sensitive regions to this flow structure under these operating conditions. Both distributions show
a significant increase of energy at $\lambda = 6.5$, where flow structures associated with $St_\Omega \in \{1,2\}$ interact mostly with the ROOT
of the blade. The dependency of $\Gamma(\varepsilon'_a)^{St_\Omega=1}$ on both FST and $\lambda$ is evidence that $St_\Omega = 1$ contributes to the aerodynamic
induced strain in the blade, as opposed to a simple contribution of gravity-induced strain dynamics.

The profiles of $\Gamma(\varepsilon'_a)^{St_\Omega=3}$ show a clear dependency on the aerodynamic behaviour of the blade. $\Gamma(\varepsilon'_a)^{St_\Omega=3}$ peaks at $\lambda_d$ for
FST cases A and B, with their relative significance increasing at $\lambda = 3.5$, and decreasing in energy under $\lambda > \lambda_d$. Furthermore,
the TIP presents the largest energy associated with this flow structure at $\lambda \in \{3.5, 4\}$, underscoring once more the relevance
of close-to-design/design operating conditions on the excitation of the blade dynamics. The distribution of $\Gamma(\varepsilon'_a)^{St_\Omega=3}$ follows
closely the profiles of $\mathrm{RMS}(\varepsilon'_a)$ presented in figure 12, with the exception of the peak of energy of $\Gamma(\varepsilon')^{St_\Omega=3}$ being observed
for $\lambda = \lambda_d$ instead of $\lambda = 3.5$. However, the combined excitation at $\lambda = 3.5$ from both $St_\Omega \in \{1,2\}$ is larger than at $\lambda = 3$,
potentially driving the increased $\mathrm{RMS}(\varepsilon'_a)$ observed at these conditions. At $\lambda > \lambda_d$, the energy is redistributed towards the
ROOT and the excitation of the TIP associated with these frequencies drops significantly in energy.

The increase of $TI$ induces a relative decrease in energy for $\Gamma(\varepsilon'_a)^{St_\Omega \in \{1,2\}}$, whilst promoting $\Gamma(\varepsilon'_a)^{St_\Omega=3}$ particularly at
off-design conditions. The distributions of $\Gamma(\varepsilon'_a)^{St_\Omega=2}$ show a particular sensitivity to FST, especially at the ROOT under
$\lambda > \lambda_d$.

Comparing $\Gamma(u')^{St_\Omega=3}$ (in figure 18 $a$)) with $\Gamma(\varepsilon')^{St_\Omega=3}$ (in figure 18 $b$)), we can see that despite that the flow structure's
energy decrease within the immediate near-wake region of the turbine as $\lambda$ increases, the blade's dynamics reflect an en-
hanced influence of the same flow structure especially at $\lambda \approx \lambda_d$ for all FST conditions. This suggests an active energy-transfer
mechanism between wake and blade, highlighting that both flow, and structural dynamics cannot be interpreted in isolation.

## 7 Conclusions and outlook

This work presents the first simultaneous measurements of wake dynamics and distributed blade strain response in a wind
turbine model under controlled free-stream turbulence (FST) and operating conditions. By integrating high-resolution Rayleigh
backscattering fibre-optic sensing with synchronized hot-wire anemometry, we have captured the spatio-temporal coupling
between coherent wake structures and blade deformation across the full operational envelope. This experimental framework
delivers resolution in both space and time, enabling direct quantification of fluid-structure interaction mechanisms that have
previously been accessible only through numerical simulations or indirect inference from separate flow, or structural analysis.

The coupled blade-wake system is governed by the complex interplay between operating condition ($\lambda$) and inflow turbu-
lence intensity. The blade's structural response exhibits distinct regimes as a function of $\lambda$. At design conditions ($\lambda = \lambda_d$),
aerodynamic loading becomes spatially uniform and dynamically coherent, yielding efficient power extraction with consid-
erable strain fluctuations. Below design operating conditions, the blade reacts mostly to low-frequency bending dynamics,



particularly at $St_\Omega \in \{1, 2\}$. Above design, while time-averaged flapwise strain continues to increase with the axial thrust acting on the rotor, the amplitude of fluctuating strain across the blade decreases, driven by a fully attached flow to the blade. We observe a peak in $\mathrm{RMS}(\varepsilon_a')$ at $\lambda \approx 3.5$—the transition point between partially stalled and design operation—suggesting that dynamic-stall conditions below near-design operation increase the aerodynamic contribution to fatigue damage.

The distributed sensing reveals pronounced spatial variation in loading mechanisms. The blade TIP consistently experiences the largest strain fluctuations (up to 75% aerodynamic-driven strain at $\lambda = \lambda_d$), observed to be dominated by tip-vortex shedding at $St_\Omega = 3$. The ROOT is primarily influenced by low-frequency dynamics ($St_\Omega = 1$), potentially driven by the wake dynamics at $St_\Omega = 1$ which are typically more energetic close to the root of the turbine relative to the tip Biswas and Buxton (2024b), combined with the influence of the presence of the tower occuring once per revolution. Meanwhile, the MIDSPAN acts as a transitional region exhibiting sensitivity to both tip and root dynamics. This spatial structure underscores that fatigue hotspots cannot be predicted from hub-integrated loads alone, and that blade design must account for section-specific excitation mechanisms.

The increase of FST intensity amplifies time-averaged strain, particularly $\Delta_f$—our proxy for aerodynamic bending moment. Larger turbulence intensity attenuates the coherence of tip vortices (reducing $\Gamma(u')^{St=3}$ in the near wake) while enhancing broadband blade excitation and increasing the energy of the coupled fluid-structure system, particularly at $St_\Omega \in \{1, 2\}$. FST effects are observed to be secondary to $\lambda$: near design operation, the sensitivity to turbulence intensity is minimized, suggesting that aerodynamic stability inherently mitigates the effects of increased inflow variability. At off-design conditions—especially $\lambda < \lambda_d$—the influence of FST is clearer, potentially by promoting flow reattachment, and modulating stall dynamics.

From an aerodynamic-induced fatigue damage perspective, our results suggest it is preferable to operate turbines slightly above design conditions rather than below, at the compromise of larger centrifugal loads. At $\lambda > \lambda_d$, aerodynamic-induced strain fluctuations stabilise, whereas at $\lambda$ immediately below $\lambda_d$, intermittent stall and reattachment dynamics increase $\mathrm{RMS}(\varepsilon_a')$, potentially leading to the more rapid accumulation of fatigue damage. Moreover, while the distributions of $\Delta_f$ closely follow typical distributions of the thrust coefficient of wind turbines, $\mathrm{RMS}(\varepsilon_a')$ follow the typical curve of $C_P$, suggesting a fundamental decoupling between time-averaged loads (thrust-driven) and fluctuating dynamics (power-driven).

Beyond these physical insights, this work establishes a new experimental paradigm for studying aeroelastic phenomena in rotating machinery. The combination of distributed fibre-optic strain sensing and synchronised wake measurements provides a data-rich framework for validating coupled fluid-structure simulations, assessing wake-steering strategies, and informing condition-based maintenance protocols. The spatial resolution achieved ($\delta f = 2.6\mathrm{mm}$ along the fibre extent) and the ability to correlate instantaneous local blade strain with local wake structure sheds light on new research directions for investigating resonance phenomena, load mitigation strategies, and the influence of atmospheric conditions on turbine longevity.

Future work will focus on exploiting the concurrent measurements of blade strain and wake velocity to further elucidate the fluid-structure coupling mechanisms. In particular, the cross-correlation and cross-power spectral density between the concurrent velocity fluctuations and the strain response along the blade will be examined to quantify the temporal and spectral coherence between flow, and structural dynamics. This analysis enables a detailed characterisation of the mechanisms governing the transfer of energy and momentum between the wake flow and the blade deformation.



In summary, this work bridges the gap between wake aerodynamics and structural dynamics by providing simultaneous, spatially resolved measurements of both fields. The results demonstrate that blade loading is not simply a function of mean inflow conditions or integrated turbine performance, but arises from a dynamic coupling between coherent flow structures and distributed structural dynamics. This understanding is essential for designing resilient, efficient wind turbines capable of operating reliably in the complex, turbulent environments characteristic of modern wind farms.

*Data availability.* Data are available upon request to the corresponding author.

*Author contributions.* F. J. G. de Oliveira was involved in formal analysis, investigation, methodology, writing—original draft and data curation. M. Bourhis was involved in draft revision and editing. Z. S. Khodaei was responsible for methodology, draft revision and supervision. O. R. H. Buxton took part in conceptualization, funding acquisition, methodology, supervision, draft revision and editing.

*Competing interests.* No competing interests are present.



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
