# Peer review of "The effect of tip-speed ratio and free-stream turbulence on the coupled wind turbine blade/wake dynamics"

_Wind Energy Science, 2025_

## Referee Comment (RC1)

Review for "The effect of tip-speed ratio and free-stream turbulence on the coupled wind turbine blade/wake dynamics" - *Francisco J. G. de Oliveira, Martin Bourhis, Zahra Sharif Khodaei, and Oliver R. H. Buxton*

This manuscript investigates the impact that tip speed ratio (TSR; 7 different TSRs tested) and free-stream turbulence (FST; 3 different FSTs tested) have on both the strain acting on a wind turbine rotor blade, and the wake of that wind turbine. To investigate the edgewise and flap-wise strain response, a Rayleigh backscattering sensor (RBS) is integrated along the span of one rotor blade in a sinusoidal pattern that allows for simultaneous measurements at multiple positions along the blade.

Overall, while the manuscript is well written, and the presentation of this set-up is very interesting, there are many open questions and some potentially severe flaws in the presentation and interpretation of the results.

The Abstract and Conclusion should be updated according to the comments.

**Major**

1. The resonance frequencies of the blades, the tower+nacelle, and the whole turbine are missing. Considering that strain fluctuations are the main focus of this study, this is crucial information to ensure that the effects are not due to resonance. Did you ensure (by measuring) that the turbine was not vibrating during the measurements?

2. One of the main aspects of this article is said to be the "coupled wind turbine blade/wake dynamics". For this, the hot-wire measurements and RBS measurements are synchronized, as described on pages 8 and 9. However, these measurements are not evaluated in a way that actually makes use of the synchronization – the hot-wire measurements and strain measurements are analyzed separately to then draw conclusions on how the blade is affected by the wake. Without actual evidence, it is e.g. concluded that the $St = 3$ – mode of the blade strain is due to a back-coupling of the tip vortex structures in the wake of the turbine farther downstream. I am doubtful about this interpretation and would rather assume that the observations are due to stronger vibrations and variations in the rotational speed (the authors report small variations but that could still be sufficient to see harmonics) at higher turbulence intensities and TSRs.
   In a similar manner, I would assume that the strain fluctuations at the blade tip are highest because the blade profiles are thinnest and most flexible there.

3. From the results presented here, I do not agree with the authors' direct conclusion that the wind turbine wake affects the blade dynamics – this to me seems like an indirect conclusion without the necessary proof, e.g. through correlating the two signals. In [1], for example, it is shown how the vibration-induced motion of a blade tip, through induced lift variations, modifies the strength of the tip vortex (not the other way round).
   Alternatively, I can see a direct argumentation where the flow field around the profile (with all the structures that are shed) does impact the blade, but that does not explain the relation of the St=3-mode to the tip vortex shedding of three blades.

4. It is assumed that the measured strain experienced by the wind turbine blade can be decomposed into gravitational, centrifugal, and aerodynamic strain-driven components, and that the impact of the former two can be obtained through calibration in a quiescent atmosphere. However,

unless these experiments were done in a vacuum, spinning the rotor would still also induce aerodynamic forces, just for different angles of attack compared to U_\infty = 2.8 m/s. The presented results do therefore not show the aerodynamic component, but *part* of the aerodynamic component. This should be corrected throughout the manuscript.

5. Error bars should be added to all figures that compare different cases (specifically 4b, 5, 11, 12, 13b, 18a+b) to allow for a proper discussion of differences.
   Also, it should be added what the errors of the hot-wire measurements are.

6. Ll. 268 "*At the same time, modifications of the flow features within the wake feed back into the structural response of the blades, as the inherent wake dynamics are imprinted on them (de Oliveira et al., 2025).*" The citation is for a cylinder in turbulent inflows and does not really match the written sentence. This should be stated clearly. Also, I am skeptical that the interpretation can be used 1:1 for a wind turbine wake, see points above.

7. Figure 11/ ll. 355: most curves do *not* show a monotonic increase, this is only the case for ROOT-C and MIDSPAN-C. Also, particularly for the TIP, without error bars, the data is too scattered to draw conclusions.

8. Figure 14/ discussion in ll. 445: The distributions at the tip are not heavy-tailed, they are pretty similar to the other positions. For better comparison, I suggest to add a Gaussian fit, preferably the same curve for all 9 sub-plots, so that deviations from a normal distribution are directly obvious.

9. Figure 17: Add the spectra for U=0 for the three TSRs.

10. Figure 18:
    a. 18a) would be better comparable with b) if it was plotted over λ, too, and the downstream distance was color-coded.
    b. For a) and b), have respectively the same axes ranges.
    c. Fig. 18b) C-St=1: the last point is not within the plot window.

11. Why are the results of case B only shown occasionally and not systematically?

**Minor**

1. References should be sorted according to their publishing year in the main part.

2. Ll. 35: The citation is for a cylinder. This should be mentioned instead of suggesting that the study was done on an airfoil. There are more appropriate citations that investigate effects of freestream turbulence on the aerodynamics of blade sections, see for example [2] and references therein, or [1,3] for an investigation of the impact of freestream turbulence on a tip vortex.

3. Ll. 60 the statement would be true for ideal systems that do not have any delay before the control is acting, however, e.g. [4] show indirectly how much λ varies in normal operation.

4. That increased FST breaks down tip vortices faster has been shown significantly earlier than 2023 (e.g., [5,6]). Also, e.g. [7-9] argue more specifically that FST accelerates the transition to fully developed turbulent wakes in general (5 for a wind turbine wake, 6 for porous disc wakes, 7 for cylinder wakes).

5. Ll. 204: do you mean that the measurement time was 120s and the synchronized measurement started after the trigger?

6. Figure 6: The color map is saturated in fig. 6a. The range should be expanded to 0 to show the homogeneity of your inflow.

7. Figure 7: The forces are typically acting on the wing's quarter point. Also, the arrow marking U_rel does not reach the corner of the rectangle spanned by the induced velocities.

8. Figure 15 and discussion: The broadening of the PDFs at $|y/R|<0.4$ could also be due to turbulence from the nacelle, and that in the shear layers due to the higher TI levels in general – in accordance with Fig. 6.
   What type of intermittency are you referring to here?
   Adding more than 0 and -2 at the y-axis would be helpful.

9. L. 401 It looks like the references *Biswas and Buxton, 2024b; Bourhis et al., 2025* are included as a reference to "tip vortices are shed from the blade tips" and not as a support for the fluctuating loads (which would be appropriate). If this is the case, I do not really see a reason for the addition here.

10. Ll. 521 Add that this is due to the interaction of tip vortices through leap-frogging and add the appropriate references (e.g., [10] or [11]).

11. Ensure that you use the same notation for quantities throughout the manuscript. There are several examples where the axes labels (e.g., L_11, ε_a) and text/captions (e.g., L, ε´_a) are different.

**Typos etc.**

1. L. 23 + corresponding reference: Porté-Agel
2. Eq. (2) $t$ and $\tau$ are not defined.
3. L. 169 its
4. L. 401 reacting more strongly
5. L. 507 $\Gamma(\varepsilon'\_a)$
6. L. 513 $\Gamma(\varepsilon'\_a)= \Gamma(\varepsilon')-\Gamma(\varepsilon'\_g)$
7. Fig. 1 – caption: please use SI units.
8. Fig. 6 $\Delta U$ undefined
9. 325 compared to

**References**

[1] Yadala S, Dehareng S, Neunaber I, et al. The effect of turbulence on a flexible finite wing: forces, deflections and the wingtip vortex. *Journal of Fluid Mechanics*. 2025;1019:A38

[2] Li L, Hearst RJ. The influence of freestream turbulence on the temporal pressure distribution and lift of an airfoil. *J Wind Eng Ind Aerodyn*. 2021; 209: 104456.

[3] Couliou, M., Yadala, S., Jankee, G. K., Neunaber, I., & Hearst, R. J. (2026) The effect of freestream turbulence on wing-tip vortex meandering and deformation. *International Journal of Heat and Fluid Flow*.117(110013).

[4] Milan, P., Wächter, M., & Peinke, J. (2013). Turbulent character of wind energy. Physical review letters, 110(13), 138701.

[5] Aubrun S, Loyer S, Hancock P, Hayden P (2013). "Wind turbine wake properties: Comparison between a non-rotating simplified wind turbine model and a rotating model." Journal of Wind Engineering and Industrial Aerodynamics, 120, 1 – 8.

[6] Maeda T, Kamada Y, Murata J, Yonekura S, Ito T, Okawa A, Kogaki T (2011). "Wind tunnel study on wind and turbulence intensity profiles in wind turbine wake." Journal of Thermal Science, 20.

[7] Neunaber, I., Hölling, M., Stevens, R. J., Schepers, G., & Peinke, J. (2020). Distinct turbulent regions in the wake of a wind turbine and their inflow-dependent locations: the creation of a wake map. *Energies*, 13(20), 5392.

[8] Vinnes, M. K., Neunaber, I., Lykke, H-M. H., & Hearst, R. J. (2023) Characterizing porous disk wakes in different turbulent inflow conditions with higher-order statistics. *Experiments in Fluids* 64, 25.

[9] Li, L., & Hearst, R. J. (2025). Effects of freestream turbulence on the wakes of circular and square cylinders. *Physical Review Fluids*, *10*(11), 114610.

[10] Lignarolo L, Ragni D, Krishnaswami C, Chen Q, ao Ferreira CS, van Bussel G (2014). "Experimental analysis of the wake of a horizontal-axis wind-turbine model." Renewable Energy, 70. Special issue on aerodynamics of offshore wind energy systems and wakes

[11] M. Felli, R. Camussi, and F. D. Felice, Mechanisms of evolution of the propeller wake in the transition and far fields, J. Fluid Mech. 682, 5 (2011)

---

## Referee Comment (RC2)

**Review of 'The effect of tip-speed ratio and free-stream turbulence on the coupled wind turbine blade/wake dynamics'**

This manuscript presents an experimental study of blade strain measurements on a wind turbine rotor using Rayleigh backscattering sensing (RBS). The authors investigate how gravity- and rotation-induced loads combine with aerodynamic loading, and propose a decomposition method to isolate centrifugal, gravitational, and aerodynamic strain contributions. Strain statistics are compared under quiescent background conditions and under controlled free-stream turbulence across a range of tip-speed ratios.

Overall, the manuscript is clearly written and the introduction provides a helpful overview of existing work and the remaining challenges in understanding the aerodynamics of rotating wind turbine blades. The experimental setup is impressive, and I believe the dataset itself could be valuable to the community. However, several key elements required to support the main conclusions are currently missing. In particular, the manuscript lacks adequate uncertainty quantification and sensor characterization, and a number of the subsequent data-processing assumptions appear insufficiently justified. As a result, some of the interpretations may not be supported by the presented evidence. I therefore recommend **major revisions** before the manuscript can be considered for publication in 'Wind Energy Science' journal.

**Major points:**

1. **Figure 1:** How homogeneous is the free-stream turbulence (FST) across the test section and across the rotor disc? Is the FST characterization reported in a separate publication? If so, this should be cited explicitly. Otherwise, additional details (mean velocity profile, turbulence intensity distribution, integral length scales, and spatial uniformity) should be provided, either in the main text or in an appendix.

2. **Section 2.2:** The uncertainty of the RBS strain measurements is not quantified. The measured strain can depend on several factors, including the bonding/gluing procedure, sensor placement, temperature effects, and surface curvature. In addition, since the study relies on time-resolved data, the dynamic response and bandwidth of the sensing chain should be documented (or cited from prior validation studies). Finally, potential asymmetry in sensor response under tensile versus compressive loading should be addressed.

3. **line 243:** *'…whereas the contribution of gravitational loads becomes negligible, potentially due to increased blade stiffening'*. This doesn't sound correct. Blade stiffening would be a valid hypothesis in case of bending moments. Here, the RBS sensor measures peak strain at $\theta=\pi$ or $0$ which implies that axial stretching of the blades due to gravity is measured. In case of such an axial loading, blade stiffening cannot reduce the $\delta\epsilon$ due to gravity. I think the reason for such an observation is different. In the blade frame, gravity acts along the chordwise and spanwise directions as a sinusoidal force whose frequency is determined by the rotational speed. In total, gravity imposes a ~10 units of strain variation. But when the blade rotates faster, the system has less time to dynamically respond to a high-frequency forcing. Therefore, the gravity effect becomes invisible in the phase-averaged measurements beyond a certain rotation speed.

4. **Figure 9**: When comparing strain signatures across TSRs, changes in the effective structural response of the blade (including centrifugal pre-tension and any shift in modal properties) should be considered. At present, the role of blade structural dynamics is not discussed, despite being potentially important for interpreting off-design behavior.

5. **Figure 11**: It is difficult to draw strong conclusions from Figure 11 without comparing against an expected bending-moment/strain distribution under steady aerodynamic loading (e.g. BEM-based prediction). Such a reference could be used for normalizing the measurement data and render results from different sections comparable.

6. **line 357**: *'The decreased rate of increase for λ > λd reflects the aerodynamic stabilization of the blade'.* I don't understand what this means. The decreased rate is most probably because of the reduced bending moment close to the blade tip, which behaves like a free end of a cantilever beam (see the previous comment).

7. **line 364**: '*This suggests that at design conditions, the effects of FST on the time-averaged loads are mitigated by the operational conditions of the turbine.'* The observation that different FST levels have limited influence on time-averaged strain at design TSR is interesting and deserves deeper discussion. The current explanation is vague. I think, the authors should elaborate mechanistically.

8. **eq 11**: The proposed method for predicting FST-related RMS fluctuations assumes uncorrelated contributions. However, the manuscript also notes unsteadiness in rotational speed; in that case, strain fluctuations can be strongly correlated with speed variations and may not be separable by the proposed approach. Simultaneous analysis of rotation-speed fluctuations and strain (e.g. conditioning, coherence analysis, or decomposition techniques such as extended POD or conditional averaging) would be required to isolate the turbulence-driven component more convincingly. The possible influence of periodic forcing at blade-passing frequency (BPF) and its harmonics should also be addressed explicitly, and maybe eliminated priorly using notch filters.

9. **line 382**: *'TIP consistently exhibits the largest fluctuation levels across all operating conditions and FST cases.'* The statement that the tip consistently exhibits the largest fluctuation levels may be expected simply because the tangential velocity (and therefore sensitivity to rotation-rate variability) increases with radius. Normalization choices and sensitivity to RPM variations should be discussed before interpreting this result as purely aerodynamic.

10. **line 385**: *'…a marked increase in overall strain fluctuations at λ ≈ 3.5 is observed, potentially emphasizing the influence of the unstable regime of partially stalled to partially attached flow conditions'* I don't think there is enough evidence not enough to support this partial stall hypothesis. I find it more probable that there's a structural natural frequency of the blade close to the frequency associated with that rotational speed.

11. **line 398**: '*Moreover, at λ ≥ λd, conditions in which the flow is attached to the blade, increased TI consistently increases RMS(ε'a) across the three sections of the blade'.* Figure 12 doesn't support this observation.

12. **line 403**: *'These results suggest that, from an aerodynamic-induced fatigue damage perspective, it is preferable to maintain wind turbines operating at slightly above design conditions with the compromise of the increased contribution of centrifugal loads, rather than slightly-below design.'* The conclusion recommending operation slightly above design TSR from a fatigue perspective appears too general. If the observed RMS behavior is influenced by the blade's structural response and/or resonance proximity, it may not generalize across turbine designs.

13. **line 428:** *'The PDFs under quiescent background conditions follow a Gaussian-like profile consistent with the periodic impact of the combined centrifugal+gravitational loads acting on the blades.'* Periodicity alone does not imply a Gaussian distribution; a purely periodic signal sampled uniformly in phase typically yields a non-Gaussian PDF. The authors should clarify the processing used and the basis for expecting Gaussian statistics.

14. **line 459:** *'The profiles of E(ε'a) are estimated from E(ε')–E(ε'g+c).'* Once again, such a decomposition does not necessarily isolate aerodynamic loading if the components are correlated or if periodic contributions remain. This may explain why the spectra remain dominated by BPF in Figure 17. A more robust separation approach should be discussed.
15. **line 514:** '…ε'a and ε'g are uncorrelated'. This comment implies that the authors disregard the results shown in figure 17. Since both components can contain rotor-synchronous periodicity, the correlation assumption should be revisited.

**Minor points:**

- **line 308:** *'…and edgewise \epsilon_a^f…'* should be \epsilon_a^e, I guess.
- **line 421:** *'Figure 15 represents…'* I guess, it should be Figure 14.